# AbdomenAtlas-8K: Annotating 8,000 CT Volumes for Multi-Organ Segmentation in Three Weeks

Chongyu Qu[1]     Tiezheng Zhang[1]     Hualin Qiao[2]     Jie Liu[3]
Yucheng Tang[4]     Alan L. Yuille[1]     Zongwei Zhou[1,*]

[1]Johns Hopkins University     [2]Rutgers University
[3]City University of Hong Kong     [4]NVIDIA

Code & Data: https://github.com/MrGiovanni/AbdomenAtlas

## Abstract

Annotating medical images, particularly for organ segmentation, is laborious and time-consuming. For example, annotating an abdominal organ requires an estimated rate of 30–60 minutes per CT volume based on the expertise of an annotator and the size, visibility, and complexity of the organ. Therefore, publicly available datasets for multi-organ segmentation are often limited in data size and organ diversity. This paper proposes an active learning procedure to expedite the annotation process for organ segmentation and creates the largest multi-organ dataset (by far) with the spleen, liver, kidneys, stomach, gallbladder, pancreas, aorta, and IVC annotated in **8,448** CT volumes, equating to **3.2 million** slices. The conventional annotation methods would take an experienced annotator up to 1,600 weeks (or roughly 30.8 years) to complete this task. In contrast, our annotation procedure has accomplished this task in three weeks (based on an 8-hour workday, five days a week) while maintaining a similar or even better annotation quality. This achievement is attributed to three unique properties of our method: (1) label bias reduction using multiple pre-trained segmentation models, (2) effective error detection in the model predictions, and (3) attention guidance for annotators to make corrections on the most salient errors. Furthermore, we summarize the taxonomy of common errors made by AI algorithms and annotators. This allows for continuous improvement of AI and annotations, significantly reducing the annotation costs required to create large-scale datasets for a wider variety of medical imaging tasks.

## 1 Introduction

Medical segmentation is a rapidly advancing task that plays a vital role in diagnosing, treating, and radiotherapy planning [37, 10, 32, 103, 40]. Building datasets of a substantial number of annotated medical images is critical for training and testing artificial intelligence (AI) models[1] [104, 52, 70]. However, medical datasets carefully annotated by an annotator are infeasible to create at a huge scale using conventional annotation methods [65] because performing per-voxel annotations is expensive and time-consuming [89, 55, 53]. As a result, publicly available datasets for multi-organ segmentation are often limited in data size (a few hundred) and organ diversity[2] as reviewed in Figure 1(a).

---

*Corresponding author: Zongwei Zhou (ZZHOU82@JH.EDU)

[1]While some of the abdominal organs can be automatically segmented by state-of-the-art AI models with fairly high accuracy, many other organs and anatomical structures remain suboptimal or unknown [55].

[2]TotalSegmentator [89] annotated a diverse set of classes for 1,024 CT volumes, but a majority of volumes were largely annotated *only* by pre-trained nnU-Nets [37]. As a result, their annotations are highly biased to the specific architecture (see §5); also, their procedure does not consider inconsistency across different AI models.

37th Conference on Neural Information Processing Systems (NeurIPS 2023) Track on Datasets and Benchmarks.

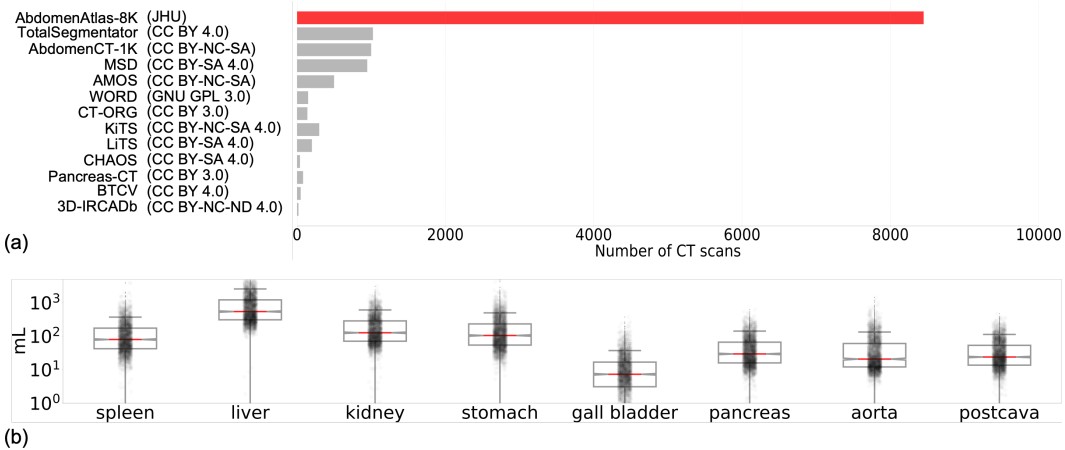

Figure 1: **(a) An overview of public datasets.** AbdomenAtlas-8K stands out from other datasets due to its large number of annotated CT volumes. We have reviewed dataset names (and licenses). **(b) Volume distribution of eight organs.** The significant variations within and across organs presented in our AbdomenAtlas-8K present challenges for the multi-organ segmentation and the generalizability of models to different domains. More comparisons can be found in Appendix Table 3 and Figure 9.

There is a pressing need to expedite the annotating procedure. The latest endeavors to construct a fully annotated dataset remained to ask radiologists to manually annotate each and every missing label [58, 36, 41, 78, 72, 9, 85, 21, 55, 99]—the same annotating procedure used a decade ago [20] or even earlier [64, 17]. Such a procedure is extremely costly, particularly for medical images, and will be time-consuming to create a large-scale dataset for every medical specialty [69, 90, 13, 83, 65]. To improve the annotation efficiency, active learning has been widely explored by combining the radiologist's competence and the computer's capability [106, 107, 71, 8, 11]. However, most studies in the active learning literature have been retrospective in nature, in which the annotating procedure was *simulated* by simply retrieving labels for the data without physically involving the radiologists in the loop to create/revise the labels [105, 66, 87, 94]. In contrast, our study is **proactive**, not only proposing a novel active learning procedure but also implementing it to actually construct a large dataset of 8,000 fully annotated CT volumes within a very short span of time, leveraging the synergy between medical professionals and AI algorithms in practice.

To overcome the deficiency of the conventional annotation method—which involves manually annotating each volume, slice, and voxel—we propose an efficient method to enable rapid organ annotation across enormous CT volumes. The efficiency has been demonstrated on 15 datasets and 8,448 abdominal CT volumes (1.2 TB in total). Our method enables high-quality, per-voxel annotations for eight organs and anatomical structures in all the CT volumes in three weeks (5 days per week; 8 hours per day), justified in §3.3. The constructed dataset is named AbdomenAtlas-8K, and its detailed statistics can be found in Figure 1. As a comparison, the conventional annotation methods [45, 26, 29, 51, 6] would take up to 1,600 weeks (30.8 years) to complete such a task[3].

Our key innovation is that rather than time-consuming annotating organs voxel by voxel, we leverage existing data and incomplete labels from an assembly of 16 public datasets. AI models are trained on the labeled part of data and generate predictions for a large number of unlabeled parts of data. We then actively revise the model predictions by only selecting the most salient part of the regions for annotators to correct. Moreover, our study provides a taxonomy of common errors made by AI algorithms and annotators. This taxonomy minimizes error duplication and augments data diversity, ensuring the sustainability of continuous revision of AI algorithms and organ annotations. Our novel active learning procedure involves only one trained annotator, three AI models, and commercial software (Pair[4]). After the procedure, annotators confirm the annotation of all the CT volumes by visual inspection. A large-scale, but private, dataset [92, 93] is used for external validation.

---

[3]Trained annotators annotate abdominal organs at a rate of 30–60 minutes per organ per three-dimensional CT volume reported by Park et al. [65]. We have also conducted a reader study to estimate the annotation time for each CT volume in our dataset, detailed in Appendix Table 4.

[4]We purchased a license from Pair. Alternatively, we also used open-source MONAI Label for annotation.

In summary, we made two major contributions. **Firstly**, AbdomenAtlas-8K was a composite dataset that unified datasets from 26 different hospitals worldwide. In total, over $60.6 \times 10^9$ voxels were annotated in comparison with $4.3 \times 10^9$ voxels annotated in the public datasets. We scaled up the organ annotation by a factor of 15 and released the masks for 5,195 of the 8,448 CT volumes. This large-scale, multi-center dataset can also impact downstream clinical tasks such as surgery, treatment, abdomen atlas, and anomaly detection. **Secondly**, the proposed active learning procedure can generate an attention map to highlight the regions to be revised by radiologists, reducing the annotation time from 30.8 years to three weeks and accelerating the annotation process by an impressive factor of 533. This strategy can quickly scale up annotations for creating many other medical imaging datasets.

## 2 Related Work

**Large dataset construction.** Kirillov et al. [43] created a huge natural image dataset of 1B masks and 11M images, but it lacks semantic information, and its efficacy is limited when applied to 3D volumetric medical images [52, 34]. Our AbdomenAtlas-8K, containing 8,448 CT volumes with per-voxel annotated eight abdominal organs, is the largest annotated CT dataset at the time this paper is written. We hereby review the existing public datasets that contained over 500 CT volumes with per-voxel annotated organs [23]. For example, AMOS [40], TotalSegmentator [89], and AbdomenCT-1K [55] provided 500, 1,024, and 1,112 annotated CT volumes, respectively. Both TotalSegmentator and AMOS derived their data from a single country, with the former reflecting the Central European population from Switzerland and the latter representing the East Asian population from China. In comparison, AbdomenAtlas-8K presented a greater data diversity because the CT volumes were collected and assembled from at least 26 different hospitals worldwide. While AbdomenCT-1K sourced data from 12 hospitals, AbdomenAtlas-8K contained approximately eight times the CT volumes (8,448 vs. 1,000) and twice the variety of annotated organs (8 vs. 4). Concurrently, we are actively expanding the classes covered by AbdomenAtlas-8K. Starting with the set of 104 classes found in TotalSegmentator, we aim to significantly diversify the range of classes covered.

**Active learning for segmentation.** *Uncertainty* and *diversity* are key criteria in active learning. Uncertainty-based criteria assess the value of annotating a data point based on the uncertainty (e.g., entropy) of AI predictions [18, 19, 56, 75, 5, 63, 11]. On the other hand, diversity-based criteria aim to select unannotated samples that differ from each other and from those already annotated [48, 25, 44, 59, 81, 82, 76]. For additional active learning methods, we refer the reader to comprehensive literature reviews [83, 62, 31, 71]; but these methods face computational complexity challenges with segmentation tasks and large unannotated data pools. To overcome this, we summarized typical errors made by humans and computers. Our active learning procedure considered the anatomical priors, uncertainty in AI prediction, and data diversity, and, importantly, pivoted a prospective application of active learning rather than retrospective studies. Moreover, the derived criteria can generate an attention map, pinpointing areas necessitating revision, thereby enabling precise detection of high-risk prediction errors (evidenced in Table 1). Consequently, our strategy markedly diminished the workload and annotation time for annotators by a factor of 533.

## 3 AbdomenAtlas-8K

**Overview.** We propose an active learning procedure comprising two components: error detection from AI predictions (§3.1) and manual revision performed by radiologists to review and edit the most significant errors detected (§3.2). By repeatedly implementing these two components, it is possible to expedite the creation of fully annotated datasets for multi-organ semantic segmentation. Finally, we describe the data construction strategies (§3.3). In this work, we have applied our approach to 8,448 CT volumes in portal (44%), arterial (37%), pre- (16%), and post-contrast (3%) phases.

### 3.1 Label Error Detection Revealed by Attention Maps

Figure 2 shows the process to generate attention maps for eight target organs. The attention maps can localize the potential error regions for human annotators to review and edit AI predictions.

**(1) Inconsistency.** To quantify inconsistency, we calculate the standard deviation of the soft predictions produced by multiple AI architectures, including Swin UNETR, nnU-Net, and U-Net. Regions

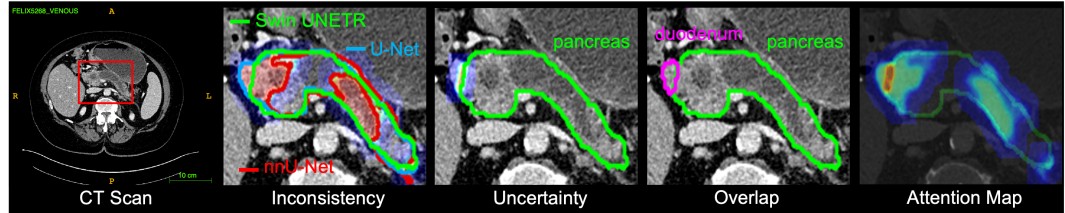

Figure 2: **Attention map generation.** The criteria inconsistency, uncertainty, and overlap refer to regions where model predictions diverge, exhibit high entropy values, and where multiple organ predictions overlap, respectively. The attention map visualizes a combination of these regions, drawing radiologists' attention to where AI predictions might falter. A standard color scheme helps in highlighting regions that merit closer review and revision. More examples are in Appendix Figure 11.

with high standard deviation indicate higher inconsistency and may require further revision.

$$\text{Inconsistency}_{i,c} = \sqrt{\frac{\sum_{n=1}^{N}(p_{i,c}^{n} - \mu_{i,c})^2}{N}}, \tag{1}$$

where the subscript $c$ represents class $c$ of our eight target organs. For each voxel $i$, $p_{i,c}^{n}$ represents the soft prediction value obtained from the $n$-th AI architecture of class $c$ at that voxel's index $i$, ranging from 0 to 1. $\mu_{i,c}$ represents the average prediction value obtained by combining the results of three AI architectures at the same voxel index. In this study, there are three AI architectures, so $N$ is equal to three. Then, the Inconsistency$_{i,c}$ value is determined by the standard deviation of the soft prediction values from the three AI architectures.

**(2) Uncertainty.** To estimate the degree of certainty linked with the AI prediction of eight target organs, we determine the entropy of the soft predictions for each organ. Regions of higher entropy values suggest diminished confidence and increased ambiguity, potentially escalating the chances of encountering prediction errors within that specific area [98], which may necessitate further revision.

$$\text{Uncertainty}_{i,c} = -\frac{\sum_{n=1}^{N} p_{i,c}^{n} \times \log(p_{i,c}^{n})}{N}. \tag{2}$$

The Uncertainty$_{i,c}$ is averaged over different AI architectures ($N = 3$).

**(3) Overlap.** The overlap in organ prediction can indicate potential errors. If a voxel is predicted to be a part of both the liver and kidneys, even without ground truth, we can reasonably forecast a prediction mistake. We use the following measure to detect organ overlap in predictions.

$$\text{Overlap}_{i,c} = \begin{cases} 1 & \text{if } p_{i,c}^{n} > 0.5 \text{ and } \exists\, p_{i,c_{\not\in}}^{n} > 0.5 \\ 0 & \text{otherwise} \end{cases} \tag{3}$$

We generate *pseudo labels* by applying a threshold of 0.5 to the probability values. Pseudo labels refer to organ labels predicted by AI models without any additional revision or validation by human annotators. The overlap value, denoted as Overlap$_{i,c}$, is determined based on the following criteria: if the prediction value for class $c$ exceeds the threshold of 0.5 for at least one AI architecture and there exists a prediction value not belonging to class $c$ that exceeds 0.5 for the same voxel index $i$, then the overlap value is set to 1; otherwise, it is set to 0.

As a result, an *attention map* is generated to help annotators quickly locate regions that require revision or confirmation. We combine the inconsistency, uncertainty, and overlapping regions to produce the attention map. Consequently, a higher Attention$_{i,c}$ value in the 3D attention map indicates a greater risk of a prediction error for that voxel.

$$\text{Attention}_{i,c} = \text{Inconsistency}_{i,c} + \text{Uncertainty}_{i,c} + \text{Overlap}_{i,c} \tag{4}$$

To assess the attention map, we identified error regions which is the summation of all false positive (FP) and false negative (FN) areas between the ground truth annotations (available in JHH [92]) and the pseudo labels predicted by AI. We then compare the attention maps with the error regions and calculate the sensitivity and precision, reported in §4.1.

## 3.2 Active Learning Procedure

**Algorithm.** Our active learning procedure has eight steps. ① Train an AI model from scratch, denoted as $\mathcal{M}_0$, using 2,100 CT volumes from 16 partially labeled public datasets. This takes approximately 40 hours. ② Direct test the current model (e.g., $\mathcal{M}_0$) on all 8,448 CT volumes to segment eight organs. This takes around 12 hours. ③ Compute organ-wise attention maps for each CT volume using the criteria of inconsistency, uncertainty, and overlap (§3.1), which highlight the regions that potentially have prediction errors and require human revision. ④ Compile a priority list sorted by the sum of attention maps. The larger the attention map is, the more urgent the CT volume requires revision. ⑤ Ask the annotators to review the top 5% (analyzed from Figure 4) CT volumes from the list and revise the pseudo labels guided by the attention maps. ⑥ Reassemble the annotation of each revised CT volume based on the label priority (detailed in the next paragraph). ⑦ Fine-tune the current model ($\mathcal{M}_t$) using the reassembled annotation to obtain $\mathcal{M}_{t+1}$. ⑧ Repeat steps ②-⑦ until the annotators confirm that the CT volume on the top of the prioritized list in step ④ does not need further revision, suggesting that AI predictions of the most important CT volumes have minimal errors.

**Technical details in ⑥: label priority.** In the assembly process, the *utmost priority* is given to the original annotations supplied by each public dataset. Subsequently, we assign *secondary priority* to the revised labels from our annotators. The pseudo labels, generated by AI models, are accorded the *lowest priority*. Based on this label priority, we reassemble the labels for each CT volume. This involves three different scenarios: inherited from the original labels of the public dataset if available, revised (if our annotator confirms that the pseudo labels have errors), and unchanged (if our annotator confirms that the pseudo labels predicted by the AI are correct). Figure 3 displays examples of the reassembled annotations from AbdomenAtlas-8K.

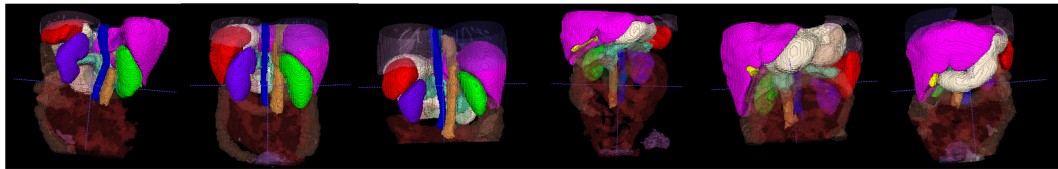

Figure 3: **AbdomenAtlas-8K annotations.** The annotations for our eight target organs (opaque) have been revised by our annotator or inherited from the original annotations of the partially labeled public datasets. The remaining organs and tumors (transparent) can be used for future research.

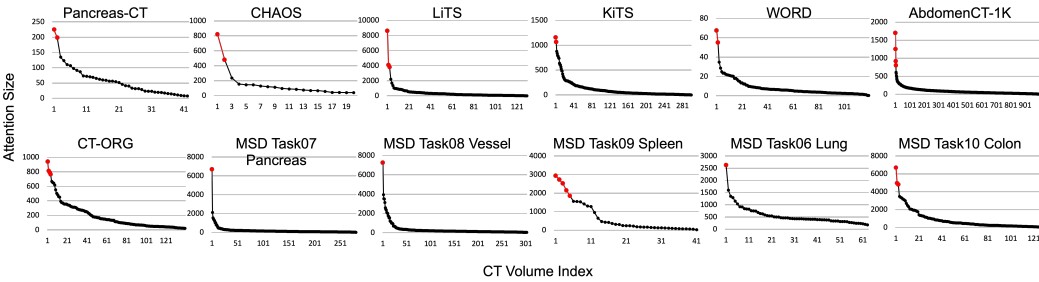

Figure 4: **Attention size distribution.** The y-axis denotes the attention size, the sum of Equation 4 over eight classes; each point corresponds to a distinct CT volume. A larger attention size implies a greater need for revision in various regions. While most CT volumes exhibit a small attention size, a few notable outliers (marked in red) stand out. These outliers are of high priority for revision by human experts. According to the figure, the ratio of outliers is about 5% (highlighted in red). The 5% is estimated by the plot and also related to the budget of human revision for each Step in the active learning procedure. It is essential to emphasize that roughly 5% of CT volumes within each dataset are highly likely to contain predicted errors, requiring further revision by our annotator.

Figure 4 illustrates the attention size of CT volumes within each partially labeled public dataset. It is important to note that the BTCV [46] and AMOS22 [40] datasets already have original annotations for our eight target organs. By analyzing the curve depicting the decreasing value of the attention map, we observed that among the 5,195 CT volumes from the 16 partially labeled abdominal datasets, only

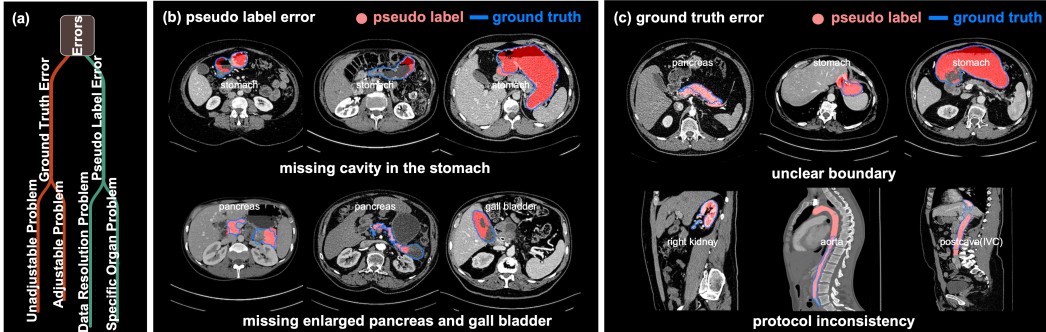

Figure 5: **(a) Error taxonomy.** To minimize repetitive errors during human revision, we collate and analyze common mistakes that occur during the data made by either AI or human annotators. **(b) Errors in pseudo labels.** For example, the cavity in the stomach is missing; the enlarged pancreas tail caused by a pancreatic tumor is missing; the enlarged gall bladder is missing. **(c) Errors in human annotations.** The high intra-annotator variability can be attributed to the ambiguity in defining organ or tumor boundaries, whereas the high inter-annotator variability is often a result of inconsistent annotation protocols and guidelines across different institutes.

5% of the volumes exhibited significantly large attention map values[5]. These high values indicate a higher risk of prediction errors, suggesting the need for further revision. Extrapolating this finding to a larger dataset of 8,000 CT volumes, we can estimate that the annotator will need to confirm and revise approximately 400 CT volumes. Assuming a rate of 15 minutes per CT volume and an 8-hour workday, this process would take approximately 12.5 days to complete. The attention map values are expected to decrease after fine-tuning $\mathcal{M}_0$ with revised labels as a new benchmark.

Figure 5(a) summarizes typical errors encountered in the active learning procedure into two categories: pseudo label errors and ground truth errors. As illustrated in Figure 5(b), pseudo label errors refer to the errors in the AI predictions. These errors often arise from irregular organ shapes, such as the absence of a cavity in the stomach or the omission of an enlarged pancreas and gall bladder. Discrepancies in CT volumes between the training and testing data, such as variations in scanners, protocols, reconstruction methods, and contrast enhancement, can also contribute to these inaccuracies. These issues result in the model's predictions lacking accuracy in representing these specific anatomical structures. As exampled in Figure 5(c), ground truth errors are errors in AI model predictions that result from inaccuracies in the human annotations used for training the model. These inaccuracies may arise due to unclear organ boundaries or inconsistency in labeling protocols across different institutions, introducing variations into the model's predictions.

## 3.3 Dataset Construction

**Annotators.** Our study recruited three annotators, comprising a senior radiologist with over 15 years of experience and two junior radiologists with three years of experience. The senior radiologist undertook the task of annotation revision in the active learning procedure. Before releasing AbdomenAtlas-8K, two junior radiologists looked through the masks in the entire AbdomenAtlas-8Kand made revisions if needed (i.e., our method missed the error regions)[6]. In addition, the two junior radiologists conducted the inter-annotator variability analysis (Figure 8) and recorded the time for conventional methods when each organ must be annotated voxel by voxel (Appendix Table 4).

**Efficiency.** *Why 30.8 years?* We considered an 8-hour workday, five days a week. A trained annotator typically needs 60 minutes per organ per CT volume [65]. Our AbdomenAtlas-8K has a total of eight organs and around 8,000 CT volumes. Therefore, annotating the entire dataset requires 60×8×8000 (minutes) / 60/8/5 = 1600 (weeks) = 30.8 (years). *Why three weeks?* Using our

---

[5]The 5% is empirically estimated based on (1) the observation in the distribution of the attention size (i.e., the number of outliers in Figure 4) and (2) the annotation budget at each Step in the active learning procedure. If there are many outliers or a limited budget, the threshold needs to be increased accordingly.

[6]Such revisions were seldom required based on our study—only 55 out of 8,448 volumes needing adjustments.

Table 1: **Evaluation metrics on JHH.** The sensitivity and precision are evaluated between our organ attention maps and organ error regions. Our attention map exhibits high average sensitivity and precision, indicating its effectiveness and accuracy in detecting false positives (FP) and false negatives (FN). This highlights its capability to precisely identify regions that need revision.

| Metrics | Spl | RKid | LKid | Gall | Liv |
|---|---|---|---|---|---|
| Sensitivity | $0.91 \pm 0.25$ | $0.93 \pm 0.17$ | $0.92 \pm 0.18$ | $0.99 \pm 0.07$ | $0.74 \pm 0.33$ |
| Precision | $0.68 \pm 0.40$ | $0.90 \pm 0.22$ | $0.85 \pm 0.23$ | $0.67 \pm 0.33$ | $0.88 \pm 0.12$ |

| Metrics | Sto | Aor | IVC | Pan | **Avg.** |
|---|---|---|---|---|---|
| Sensitivity | $0.98 \pm 0.10$ | $0.98 \pm 0.11$ | $0.98 \pm 0.09$ | $0.90 \pm 0.21$ | $0.93 \pm 0.17$ |
| Precision | $0.85 \pm 0.16$ | $0.82 \pm 0.21$ | $0.75 \pm 0.22$ | $0.91 \pm 0.15$ | $0.81 \pm 0.22$ |

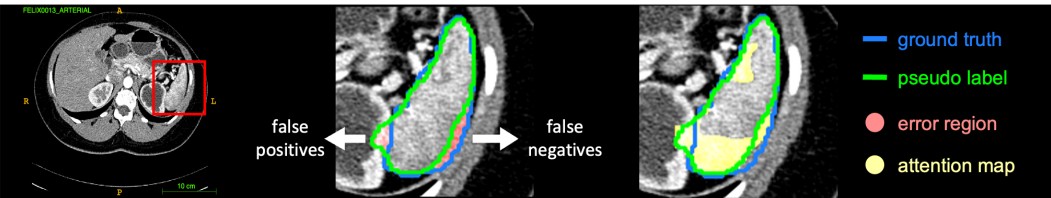

Figure 6: **Error region vs. attention map.** The discrepancy between the model's pseudo labels and the ground truth delineates the error regions in predictions, comprising both false positives (FP) and false negatives (FN). These error regions serve as the benchmark for evaluating the sensitivity and precision of our attention map (§4.1).

active learning strategy, only 400 CT volumes require manual revision from the human annotator (15 minutes per volume). That is, we managed to accelerate the annotation process by a factor of $60 \times 8/15 = 32$ per CT volume. Therefore, we completed the entire annotation procedure within three weeks, as reported in the paper. Human efforts: $400 \times 15$ (minutes) / $60/8 = 12.5$ (days) plus the time commitment for training and testing AI models taking approximately 8.5 (days).

**Dataset splits.** For 8,448 CT volumes in AbdomenAtlas-8K, we split them into training (5500 CT volumes), validation (500), and test (2448) sets. Each volume contains per-voxel annotations of the spleen, liver, left& right kidneys, stomach, gallbladder, pancreas, aorta, and IVC. Note that the JHH dataset is a proprietary, multi-resolution, multi-phase collected from Johns Hopkins Hospital [92, 42, 100, 47]. This dataset is used for external validation, including the assessment of the attention map quality (Table 1), AI generalizability (Table 2), the improvement of label quality in the active learning procedure (Appendix Table 5), and the performance of AI trained on a combination of public datasets and our AbdomenAtlas-8K(Appendix Table 6).

## 4 Experiment & Result

### 4.1 Attention Map and Annotation Evaluation

**Evaluation of attention map**. Our attention map is evaluated on 1,000 CT volumes of the JHH dataset [92], which is not included in the training process. The evaluation is performed across the eight target organs. Once an error is detected, it counts as a hit; otherwise, as a miss. To evaluate the quality of the attention map, two metrics are used: Sensitivity = TP / (TP + FN) and precision = TP / (TP + FP), where a true positive (TP) means the attention map found real mistakes the AI made; a false negative (FN) means it missed some of the AI's mistakes; and a false positive (FP) means it found mistakes where the AI was actually right. Sensitivity and precision are all calculated at the volume (a group of voxels) level rather than the voxel level. We chose sensitivity and precision because this experiment is designed to evaluate an error detection task rather than a segmentation task (comparing the boundary of attention maps and error regions). They can measure how well the attention maps detect the real error regions and whether the errors being detected are real errors, respectively. The results of the attention map evaluation are presented in Table 1. The mean values of sensitivity and precision metrics for our attention maps, with respect to eight target organs, are $0.93 \pm 0.17$ and $0.81 \pm 0.22$. Figure 6 shows the error regions and our attention map. These results

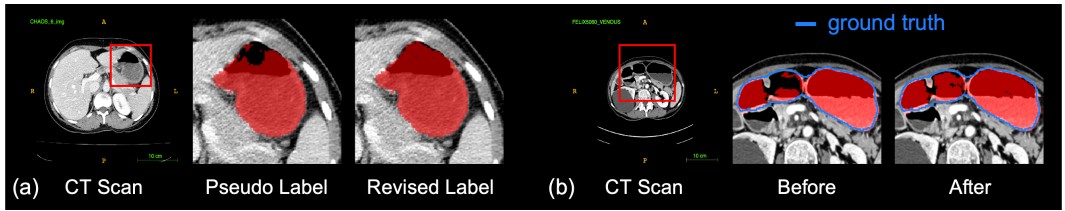

Figure 7: **(a) Pseudo vs. revised labels.** Pseudo labels are predicted by AI and then revised by annotators based on attention maps, resulting in revised labels. **(b) AI predictions before vs. after fine-tuning.** The post-fine-tuning results exhibit superior accuracy in segmenting the organ compared to the pre-fine-tuning result. Appendix Table 5 quantifies the DSC scores before and after fine-tuning.

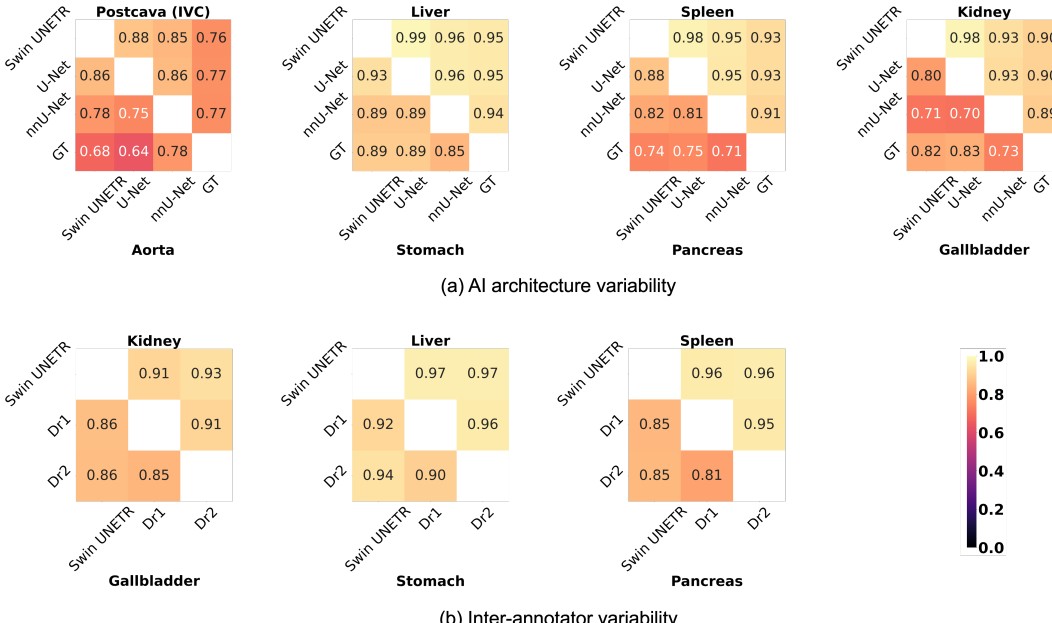

Figure 8: The DSC score in each cell is calculated between the corresponding row and column. **(a) AI architecture variability.** The segmentation predictions made by different AI architectures display minor variations for eight organs, as evidenced by the corresponding DSC scores. Overall, the AI predictions align closely with the annotations provided by human experts. **(b) Inter-annotator variability.** The DSC scores between AI predictions and human annotators consistently outperform the score between the two human annotators, suggesting a comparable annotation quality between AI predictions and human annotations in the segmentation of six organs.

suggest that our attention map effectively captures the false positives and false negatives in error regions, demonstrating a high degree of accuracy in identifying prediction errors.

**Evaluation of human annotation quality.** We provided the visualization of pseudo labels (generated by AI) and revised labels (annotated by humans) in Figure 7(a). The revised labels were used to fine-tune the AI; we then compared the pseudo labels predicted by the AI before and after fine-tuning for the same CT volume from the JHH dataset, as shown in Figure 7(b). After fine-tuning the AI using the revised labels, the AI was able to accurately segment the entire stomach on the previously unseen CT volumes. This demonstrated the high quality of the revised labels and their efficacy in enhancing AI performance. We further quantified the human annotation quality measured by Dice Similarity Coefficient (DSC) and normalized surface Dice (NSD) using 1,000 CT volumes from the JHH dataset. The results in Appendix Table 5 show continual improvements in label quality along the active learning procedure. For example, AI models exhibited a marked improvement in the (pseudo) labels of aorta and postcava, jumping from 72.3% to 83.7% and 76.1% to 78.6%, respectively.

## 4.2 Label Bias and Segmentation Quality Evaluation

**Evaluation of label bias.** We first evaluate the predictions made by three AI architectures, i.e., Swin UNETR, nnU-Net, and U-Net, on CT volumes derived from the JHH dataset [92]. The segmentation predictions of these architectures are illustrated in Appendix Figure 10. We use the original annotations of the JHH dataset [92], which include our eight target organs. Consequently, for each CT volume, we have four predictions: three from the AI models and one from human experts. We then compute the DSC score between each pair across eight organs. These results are presented in Figure 8(a) and demonstrate that while the three AI architectures produce slightly varied predictions, they closely resemble those performed by human experts. To prevent label bias to a specific architecture, the final annotations in our AbdomenAtlas-8K are determined by averaging the three AI predictions. Our approach contrasts with the TotalSegmentator dataset [89], which exclusively relies on nnU-Net for segmentation. This could potentially generate a biased dataset that impedes model generalization across different architectures or scenarios.

**Evaluation of AI prediction quality.** To assess the automated annotation quality in AbdomenAtlas-8K, we asked for the assistance of two additional human annotators to help us modify the pseudo labels of our target eight organs. Due to time limitations, they are only able to revise six out of the eight organs, specifically on the 17 CT volumes from the BTCV dataset [46]. We compute DSC scores comparing AI predictions with those annotated independently by two human annotators, referred to as Dr1 and Dr2. Figure 8(b) presents the results for six organs. The DSC scores between AI predictions and each human annotator (Swin UNETR vs. Dr1 or Swin UNETR vs. Dr2) consistently exceed the score between the two human annotators (Dr1 vs. Dr2). These findings suggest that the automated AI annotations fall within the range of inter-annotator variability, thereby indicating that the quality of our automated annotations is comparable to human annotations.

## 4.3 Benchmarking

AbdomenAtlas-8K enables precision medicine for various downstream applications. We showcased one of the most pressing applications—early detection and localization of pancreatic cancer, an extremely deadly disease with a 5-year relative survival rate of only 12% in the United States. The AI trained on a large, private dataset at Johns Hopkins Hospital (JHH) performed arguably higher than typical radiologists [92, 47, 42]. But this AI model and annotated dataset were inaccessible due to the many policies. Now, our paper demonstrated that using AbdomenAtlas-8K (100% made up of publicly accessible CT volumes), AI can achieve similar performance when directly tested on the JHH dataset (see Table 2). This study is a concrete demonstration of how AbdomenAtlas-8K can be used to train AI models that can be generalized to many CT volumes from novel hospitals and be adapted to address a range of clinical problems. For a more in-depth analysis of the segmentation performance of AI models, we present the comprehensive category-wise scores comparison across eight organs in Appendix Table 7.

Table 2: **Benchmark results of AI models trained on AbdomenAtlas-8K.** We directly apply four AI models, i.e. SwinUNETR, UNETR, U-Net, and SegResNet, trained on AbdomenAtlas-8K (public) to JHH (unseen, private) and compared its performance with AI trained on JHH. The AI models trained on AbdomenAtlas-8K exhibit comparable results in average mDSC and mNSD across eight organs when compared with AI models trained on the JHH dataset, proving the high generalization capacity of AI models trained on AbdomenAtlas-8K.

| AI Models | Trained on JHH (private) | | Trained on AbdomenAtlas-8K (public) | |
| --- | --- | --- | --- | --- |
| | mDSC (%) | mNSD (%) | mDSC (%) | mNSD (%) |
| SwinUNETR [84] | 84.8 ± 12.6 | 66.5 ± 14.8 | 86.5 ± 7.5 | 60.8 ± 10.9 |
| UNETR [28] | 78.6 ± 13.0 | 53.9 ± 14.1 | 86.6 ± 6.5 | 59.5 ± 10.4 |
| U-Net [73] | 84.8 ± 11.2 | 64.2 ± 14.8 | 87.3 ± 6.0 | 61.0 ± 10.2 |
| SegResNet [12] | 87.6 ± 6.60 | 64.9 ± 10.9 | 87.0 ± 6.1 | 60.4 ± 10.1 |

## 5 Discussion

**Impact.** The scientific community has generally agreed that large volumes of annotated data are required for developing effective AI algorithms [39, 2, 60, 24, 27, 101]. For example, developing

Foundation Models for healthcare has recently raised much attention. The Foundation Model refers to an AI model that is trained on a large dataset and can be adapted to many specific downstream applications. This requires a large-scale, fully-annotated dataset. The currently available medical datasets are too small to represent the real data distribution in clinics [80, 13, 68]. The availability of large-scale, multi-center, fully-labeled data (summarized in Appendix Table 3) is one of the most significant cornerstones for both the development and evaluation of CADe systems [90, 103]. In recent years, the rise of imaging data archives [74, 57, 38, 3, 14, 16, 58, 97, 49, 99] and international competitions [45, 86, 78, 79, 30, 29, 1, 15, 4, 40] produced several publicly available datasets for benchmarking AI algorithms, but these datasets usually are of small size, contain partial labels, come from various scanners & protocols, and are therefore often limited in their scope [22, 50, 102, 95, 96, 77, 67, 88, 42]. We anticipate that our AbdomenAtlas-8K can play an important role in enabling the model to capture complex organ patterns, variations, and features across different imaging phases, modalities, and a wide range of populations. This has been partially evidenced in Appendix Table 6, wherein we generalize AI to CT volumes taken from different hospitals.

**Limitation.** Pseudo-labels have the potential to expedite tumor annotation procedures as well, but the risk of producing a large number of false positives is a significant concern. The presence of false positives could significantly increase the time annotators need to accept or reject a detection. In total, our AI models generate 51,852 tumor masks, including tumors in the colon, liver, hepatic vessel, pancreas, kidneys, and lung. However, only using AbdomenAtlas-8K, we are not able to assess the AI performance of tumor detection due to the lack of comprehensive pathology reports or expert annotations to describe the tumors in the majority of these publicly available CT volumes. To address this problem, we collected 392 tumor-free *(private)* CT volumes from Johns Hopkins Hospital to evaluate the false positives in the pseudo labels. Using kidney tumor detection as an example, there are 37 out of 392 CT volumes containing false positives (FPR = 9.4%) and a total of 161 false positives in the 37 CT volumes. Furthermore, we stratified the dataset based on the type of blood vessels, identifying average false positive rates of 11.83% and 4.6% for the venous and arterial vasculature, respectively. Therefore, while pseudo labels can be helpful, we anticipate a more effective method for false positive reduction is needed to enable a clinically practical tumor annotation, given the complexity of tumors compared with organs. As an extension, we plan to add tumor annotations to AbdomenAtlas-8K in three possible directions. **Firstly**, we plan to recruit more experienced radiologists to revise tumor annotations. **Secondly**, we will incorporate the pathology reports (based on biopsy results) into the human revision. These actions can reduce potential label biases and label errors from human annotators. **Thirdly**, we will exploit synthetic data (tumors) that can produce enormous tumor examples and their precise masks for AI training and validation [32, 47, 33].

In TotalSegmentator, the labels were largely generated by a single nnU-Net re-trained continually. Depending solely on nnU-Net could introduce a potential label bias favoring the nnU-Net architecture. This means that whenever TotalSegmentator is employed for benchmarking, nnU-Net would always outperform other segmentation architectures (e.g., UNETR, TransUNet, SwinUNETR, etc.). This observation has been made in several publications such as Huang et al. [35]. In contrast, our AbdomenAtlas-8K incorporates predictions from three different AI architectures, preventing bias towards one specific architecture [89]. However, such a solution comes with increased computational costs, and the performance of the AI architectures can vary. Taking an average of the predictions may result in the final outcome being pulled down by poorly performing AI architectures.

## 6 Conclusion

Our study shows the effectiveness of an active learning procedure, which combines the expertise of an annotator with the capabilities of a trained AI model. It not only deploys multiple AI models to detect prediction errors but also prompts annotators to revise these potential failures. Using this approach, we successfully annotated 8,448 abdominal CT volumes of different organs within three weeks, expediting the annotation process by a staggering factor of 533. Furthermore, our experiments demonstrate that the interaction between human and AI models can produce comparable or even superior results than that of a human annotator alone. This indicates that our approach can leverage the strengths of both human and AI models to achieve accurate and efficient annotation. Leveraging our efficient annotation framework, we anticipate that larger-scale medical datasets of various modalities, organs, and abnormalities can be curated at a significantly reduced cost, ultimately contributing to the development of Foundation Models in the medical domain [91, 52, 61].

## Acknowledgments and Disclosure of Funding

This work was supported by the Lustgarten Foundation for Pancreatic Cancer Research and the Patrick J. McGovern Foundation Award. We appreciate the effort of the MONAI Team to provide open-source code for the community. This work has partially utilized the GPUs provided by ASU Research Computing. We thank Elliot K. Fishman, Linda Chu, and Satomi Kawamoto for providing the JHH dataset for external validation; Yuxiang Lai for generating the 3D rendering of segmentation; thank Xiaoxi Chen for reviewing and revising AI predictions; thank Seth Zonies and Andrew Wichmann for providing legal advice on the release of AbdomenAtlas-8K; thank Yu-Cheng Chou, Jieneng Chen, Junfei Xiao, Wenxuan Li, and Xiaoding Yuan for their constructive suggestions at several stages of the project. The content and dataset of this paper are covered by patents pending.

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
