# Appendix

## A    Implementation Details

Our AbdomenAtlas-8K includes three different AI architectures: Swin UNETR, U-Net, and nnU-Net. Swin UNETR and U-Net are trained on 2,100 CT volumes from 16 partially labeled public abdominal datasets (details in Figure 1). We use the Adam optimizer with an initial learning rate of $1e$-4, which is dynamically adjusted based on the combined dice and BCE losses. The weight decay is set to $1e$-5. nnU-Net is trained on 47 CT volumes from the BTCV [46] dataset. We utilize the SGD optimizer with an initial learning rate of $1e$-2 and a weight decay of $3e$-5. The AI architectures are trained on portal venous CT volumes. Eight NVIDIA RTX Quadro 8000 GPUs are used, and the batch size is 4 per GPU. The training process takes approximately 40 hours and the testing process takes around 12 hours.

## B    Datasets and Permissions

We will only disseminate the annotations of the CT volumes separately, and users will retrieve the original CT volumes, if needed, from the original sources (websites). Everything we intend to create and license-out will be in separate files, and no modifications are necessary to the original CT volumes. The source and permissions are elaborated in Table 3. We have consulted with the lawyers at Johns Hopkins University, confirming the permissions of distributing the annotations based on the license of each dataset.

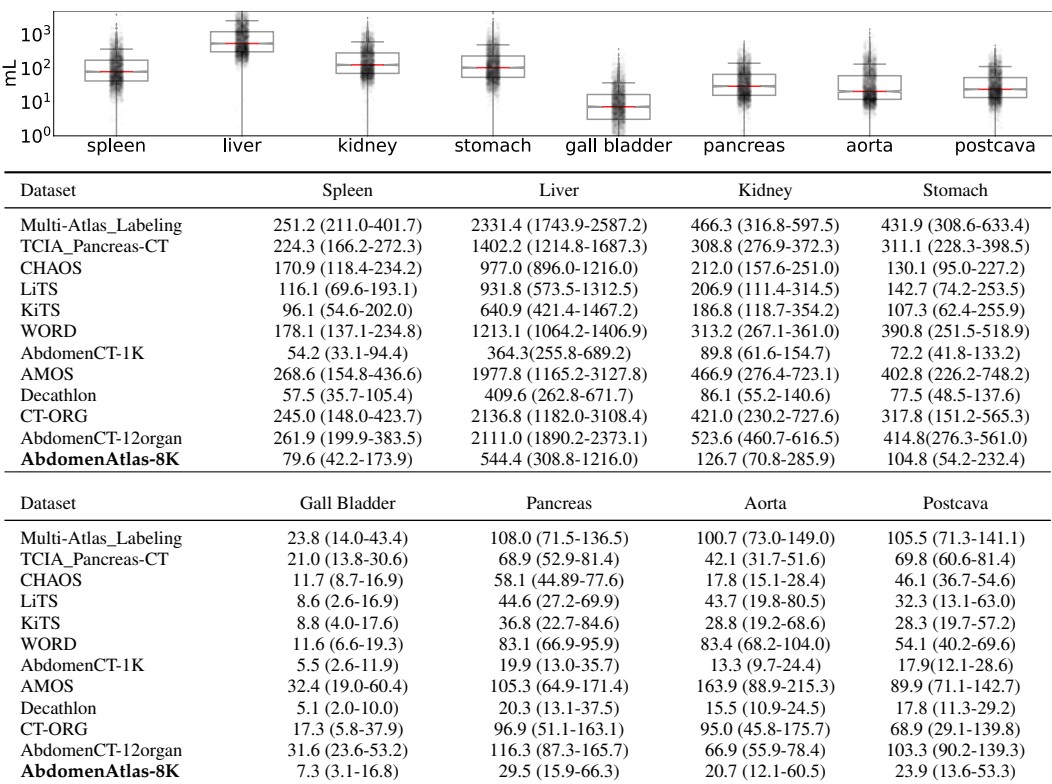

| Dataset | Spleen | Liver | Kidney | Stomach |
|---|---|---|---|---|
| Multi-Atlas_Labeling | 251.2 (211.0-401.7) | 2331.4 (1743.9-2587.2) | 466.3 (316.8-597.5) | 431.9 (308.6-633.4) |
| TCIA_Pancreas-CT | 224.3 (166.2-272.3) | 1402.2 (1214.8-1687.3) | 308.8 (276.9-372.3) | 311.1 (228.3-398.5) |
| CHAOS | 170.9 (118.4-234.2) | 977.0 (896.0-1216.0) | 212.0 (157.6-251.0) | 130.1 (95.0-227.2) |
| LiTS | 116.1 (69.6-193.1) | 931.8 (573.5-1312.5) | 206.9 (111.4-314.5) | 142.7 (74.2-253.5) |
| KiTS | 96.1 (54.6-202.0) | 640.9 (421.4-1467.2) | 186.8 (118.7-354.2) | 107.3 (62.4-255.9) |
| WORD | 178.1 (137.1-234.8) | 1213.1 (1064.2-1406.9) | 313.2 (267.1-361.0) | 390.8 (251.5-518.9) |
| AbdomenCT-1K | 54.2 (33.1-94.4) | 364.3(255.8-689.2) | 89.8 (61.6-154.7) | 72.2 (41.8-133.2) |
| AMOS | 268.6 (154.8-436.6) | 1977.8 (1165.2-3127.8) | 466.9 (276.4-723.1) | 402.8 (226.2-748.2) |
| Decathlon | 57.5 (35.7-105.4) | 409.6 (262.8-671.7) | 86.1 (55.2-140.6) | 77.5 (48.5-137.6) |
| CT-ORG | 245.0 (148.0-423.7) | 2136.8 (1182.0-3108.4) | 421.0 (230.2-727.6) | 317.8 (151.2-565.3) |
| AbdomenCT-12organ | 261.9 (199.9-383.5) | 2111.0 (1890.2-2373.1) | 523.6 (460.7-616.5) | 414.8(276.3-561.0) |
| **AbdomenAtlas-8K** | 79.6 (42.2-173.9) | 544.4 (308.8-1216.0) | 126.7 (70.8-285.9) | 104.8 (54.2-232.4) |

| Dataset | Gall Bladder | Pancreas | Aorta | Postcava |
|---|---|---|---|---|
| Multi-Atlas_Labeling | 23.8 (14.0-43.4) | 108.0 (71.5-136.5) | 100.7 (73.0-149.0) | 105.5 (71.3-141.1) |
| TCIA_Pancreas-CT | 21.0 (13.8-30.6) | 68.9 (52.9-81.4) | 42.1 (31.7-51.6) | 69.8 (60.6-81.4) |
| CHAOS | 11.7 (8.7-16.9) | 58.1 (44.89-77.6) | 17.8 (15.1-28.4) | 46.1 (36.7-54.6) |
| LiTS | 8.6 (2.6-16.9) | 44.6 (27.2-69.9) | 43.7 (19.8-80.5) | 32.3 (13.1-63.0) |
| KiTS | 8.8 (4.0-17.6) | 36.8 (22.7-84.6) | 28.8 (19.2-68.6) | 28.3 (19.7-57.2) |
| WORD | 11.6 (6.6-19.3) | 83.1 (66.9-95.9) | 83.4 (68.2-104.0) | 54.1 (40.2-69.6) |
| AbdomenCT-1K | 5.5 (2.6-11.9) | 19.9 (13.0-35.7) | 13.3 (9.7-24.4) | 17.9(12.1-28.6) |
| AMOS | 32.4 (19.0-60.4) | 105.3 (64.9-171.4) | 163.9 (88.9-215.3) | 89.9 (71.1-142.7) |
| Decathlon | 5.1 (2.0-10.0) | 20.3 (13.1-37.5) | 15.5 (10.9-24.5) | 17.8 (11.3-29.2) |
| CT-ORG | 17.3 (5.8-37.9) | 96.9 (51.1-163.1) | 95.0 (45.8-175.7) | 68.9 (29.1-139.8) |
| AbdomenCT-12organ | 31.6 (23.6-53.2) | 116.3 (87.3-165.7) | 66.9 (55.9-78.4) | 103.3 (90.2-139.3) |
| **AbdomenAtlas-8K** | 7.3 (3.1-16.8) | 29.5 (15.9-66.3) | 20.7 (12.1-60.5) | 23.9 (13.6-53.3) |

Figure 9: **The comparisons of volume distribution between public datasets and our dataset.** The values, expressed in mL, represent the median of the corresponding organ in the dataset, with the 25th and 75th percentiles indicated in parentheses.

Table 3: **The information for an assembly of datasets.** After summarizing the existing public datasets, we found that a combination of these datasets is fairly large (a total of 8,000 CT volumes and many more are coming over the years). The main challenge, however, is the absence of comprehensive annotations. In fact, scaling up annotations is much harder than scaling up CT volumes due to the limited time of expert radiologists. The existing labels of the public datasets are partial and incomplete, e.g., LiTS only had labels for the liver and liver tumors, and KiTS only had labels for the kidneys and kidney tumors. Our AbdomenAtlas-8K offered detailed annotations for all eight organs within each CT volume. With this, our contribution is significant.

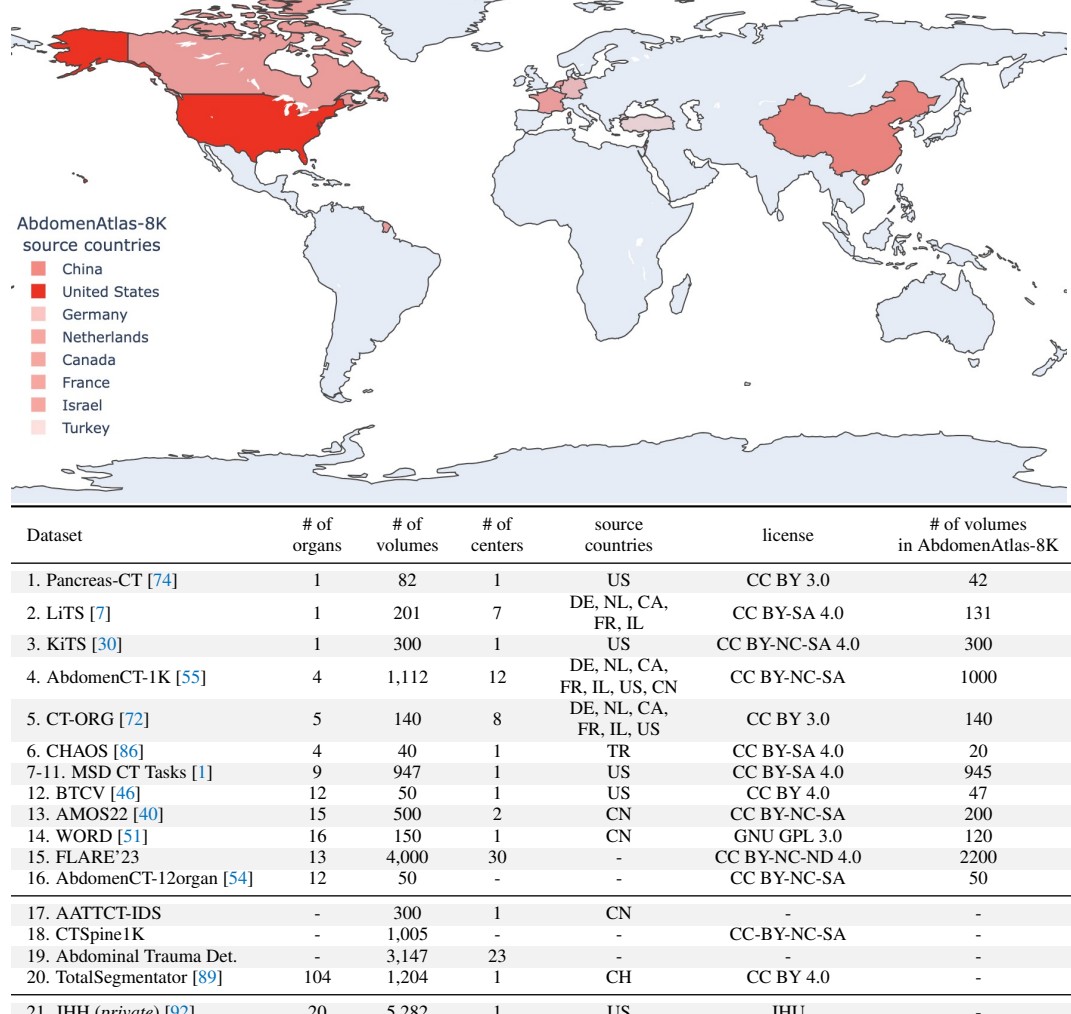

| Dataset | # of organs | # of volumes | # of centers | source countries | license | # of volumes in AbdomenAtlas-8K |
|---|---|---|---|---|---|---|
| 1. Pancreas-CT [74] | 1 | 82 | 1 | US | CC BY 3.0 | 42 |
| 2. LiTS [7] | 1 | 201 | 7 | DE, NL, CA, FR, IL | CC BY-SA 4.0 | 131 |
| 3. KiTS [30] | 1 | 300 | 1 | US | CC BY-NC-SA 4.0 | 300 |
| 4. AbdomenCT-1K [55] | 4 | 1,112 | 12 | DE, NL, CA, FR, IL, US, CN | CC BY-NC-SA | 1000 |
| 5. CT-ORG [72] | 5 | 140 | 8 | DE, NL, CA, FR, IL, US | CC BY 3.0 | 140 |
| 6. CHAOS [86] | 4 | 40 | 1 | TR | CC BY-SA 4.0 | 20 |
| 7-11. MSD CT Tasks [1] | 9 | 947 | 1 | US | CC BY-SA 4.0 | 945 |
| 12. BTCV [46] | 12 | 50 | 1 | US | CC BY 4.0 | 47 |
| 13. AMOS22 [40] | 15 | 500 | 2 | CN | CC BY-NC-SA | 200 |
| 14. WORD [51] | 16 | 150 | 1 | CN | GNU GPL 3.0 | 120 |
| 15. FLARE'23 | 13 | 4,000 | 30 | - | CC BY-NC-ND 4.0 | 2200 |
| 16. AbdomenCT-12organ [54] | 12 | 50 | - | - | CC BY-NC-SA | 50 |
| 17. AATTCT-IDS | - | 300 | 1 | CN | - | - |
| 18. CTSpine1K | - | 1,005 | - | - | CC-BY-NC-SA | - |
| 19. Abdominal Trauma Det. | - | 3,147 | 23 | - | - | - |
| 20. TotalSegmentator [89] | 104 | 1,204 | 1 | CH | CC BY 4.0 | - |
| 21. JHH (*private*) [92] | 20 | 5,282 | 1 | US | JHU | - |

US: United States    DE: Germany    NL: Netherlands    CA: Canada    FR: France    IL: Israel
CN: China    TR: Turkey    CH: Switzerland

Table 4: **The annotation time.** We employed two trained annotators to produce full annotations for the eight organs and recorded the average time needed for each organ. Both annotators have over 3-year experience in radiological image analysis.

| Organs | Annotator #1 | Annotator #2 | Average |
|---|---|---|---|
| Spleen | 30 min | 28 min | 29 min |
| Right Kidney | 26 min | 28 min | 27 min |
| Left Kidney | 23 min | 25 min | 24 min |
| Gall Bladder | 11 min | 10 min | 10 min |
| Liver | 40 min | 40 min | 40 min |
| Stomach | 32 min | 32 min | 32 min |
| Aorta | 22 min | 20 min | 21 min |
| Postcava (IVC) | 27 min | 30 min | 28 min |
| Pancreas | 19 min | 19 min | 19 min |

Table 5: **Label quality improvement along the active learning procedure.** Assessing label quality is integral to ensure that the iterative training process is harnessing the most informative samples and that the revised labels are improved in quality. In doing so, we executed an external validation by leveraging a distinct dataset (JHH) separate from the active learning pool. Specifically, we trained AI models on the dataset at different stages: (1) before revising, (2) after revising Step #1, and (3) after revising Step #2. Subsequently, these three AI models were evaluated on the unseen JHH dataset. We randomly sampled 1,000 CT volumes from JHH and computed the mean DSC scores (mDSC) and normalized surface dice (mNSD) with a tolerance of 1mm for eight organs. After label revisions, both mDSC and mNSD showed an increase, demonstrating the effectiveness of the revised organ labels. Furthermore, we highlighted the two most significantly revised classes at each step in blue. For these organs, such as the aorta and postcava, AI models exhibited a marked improvement, jumping from 72.3% to 83.7% and 76.1% to 78.6%, respectively. Conversely, the segmentation performance for other classes saw only a slight uptick. This can be attributed to two key reasons: firstly, certain organs underwent minimal revisions at given steps, and secondly, some segmentation outcomes had already reached an apex of performance, exemplified by scores like DSC exceeding 90%.

| Organ | Before revising | | After revising Step #1 | | After revising Step #2 | |
|---|---|---|---|---|---|---|
| | mDSC (%) | mNSD (%) | mDSC (%) | mNSD (%) | mDSC (%) | mNSD (%) |
| Spleen | $94.0 \pm 7.6$ | $71.6 \pm 10.5$ | $94.1 \pm 7.2$ | $71.8 \pm 11.5$ | $94.7 \pm 5.5$ | $74.8 \pm 9.5$ |
| Right Kidney | $91.9 \pm 8.0$ | $64.8 \pm 10.0$ | $92.1 \pm 7.0$ | $65.0 \pm 10.1$ | $92.5 \pm 7.0$ | $64.7 \pm 7.7$ |
| Left Kidney | $90.8 \pm 8.7$ | $65.3 \pm 10.1$ | $91.0 \pm 7.9$ | $66.2 \pm 10.1$ | $91.5 \pm 7.4$ | $64.9 \pm 9.8$ |
| Gall Bladder | $82.4 \pm 14.3$ | $59.2 \pm 17.2$ | $79.8 \pm 15.9$ | $54.3 \pm 17.6$ | $81.7 \pm 12.4$ | $55.6 \pm 15.1$ |
| Liver | $95.5 \pm 6.2$ | $64.3 \pm 9.5$ | $95.7 \pm 4.9$ | $63.9 \pm 9.2$ | $95.8 \pm 4.4$ | $62.8 \pm 8.2$ |
| Stomach | $91.5 \pm 8.0$ | $52.7 \pm 10.3$ | $91.9 \pm 6.8$ | $53.9 \pm 10.2$ | $92.4 \pm 5.6$ | $53.0 \pm 8.8$ |
| Aorta | $72.3 \pm 12.1$ | $55.6 \pm 11.3$ | $82.9 \pm 10.2$ | $63.8 \pm 11.5$ | $83.7 \pm 8.8$ | $64.2 \pm 9.8$ |
| Postcava (IVC) | $76.1 \pm 12.0$ | $52.3 \pm 11.0$ | $77.2 \pm 11.0$ | $63.6 \pm 10.7$ | $78.6 \pm 10.1$ | $54.5 \pm 9.5$ |
| Pancreas | $79.4 \pm 12.4$ | $51.5 \pm 10.1$ | $80.1 \pm 11.0$ | $52.6 \pm 9.9$ | $79.2 \pm 12.4$ | $51.6 \pm 9.3$ |
| Average | $86.0 \pm 9.9$ | $59.7 \pm 11.1$ | $87.2 \pm 9.1$ | $60.6 \pm 11.2$ | $87.8 \pm 8.2$ | $60.7 \pm 9.7$ |

Table 6: **Assessing AI generalizability to multiple novel datasets.** On the left panel, the AI was trained on 16 partially labeled datasets; on the right panel, the AI was trained on the subset of AbdomenAtlas-8K. Note that the subset of AbdomenAtlas-8K has the same number of CT volumes (5,195), but their annotations are more comprehensive than 16 public datasets. We evaluated the segmentation performance of the two AI models on three unseen datasets with different sample sizes ($N$), i.e., FLARE'23 [53] ($N = 300$), TotalSegmentator [89] ($N = 1,204$) and JHH [92] ($N = 5,282$). The performance is measured by mean DSC scores (mDSC) and mean normalized surface dice (mNSD) with a tolerance of 1mm. The results showed that when pre-training the AI model using AbdomenAtlas-8K, the average mDSC and mNSD scores improved. This suggests that AbdomenAtlas-8K has higher quality compared to partially labeled abdominal datasets.

| **FLARE'23** [53] | Trained on 16 partially labeled datasets | | Trained on AbdomenAtlas-8K | |
|---|---|---|---|---|
| | mDSC (%) | mNSD (%) | mDSC (%) | mNSD (%) |
| Spleen | $96.1 \pm 1.9$ | $82.0 \pm 16.4$ | $96.4 \pm 1.9^{****}$ | $84.7 \pm 16.6^{****}$ |
| Right Kidney | $94.1 \pm 6.7$ | $81.5 \pm 12.2$ | $96.3 \pm 1.9^{***}$ | $80.0 \pm 14.1^{**}$ |
| Left Kidney | $93.9 \pm 9.2$ | $81.3 \pm 12.7$ | $93.1 \pm 9.7^{ns}$ | $79.5 \pm 14.6^{****}$ |
| Gall Bladder | $85.1 \pm 5.0$ | $50.5 \pm 16.3$ | $86.0 \pm 3.9^{***}$ | $57.2 \pm 13.0^{****}$ |
| Liver | $97.3 \pm 1.2$ | $78.7 \pm 15.8$ | $96.8 \pm 2.0^{****}$ | $76.6 \pm 14.9^{***}$ |
| Stomach | $91.8 \pm 5.0$ | $49.2 \pm 16.4$ | $91.5 \pm 6.3^{ns}$ | $50.0 \pm 17.3^{ns}$ |
| Aorta | $83.7 \pm 14.7$ | $72.0 \pm 18.4$ | $81.8 \pm 17.8^{ns}$ | $73.4 \pm 16.7^{ns}$ |
| Postcava (IVC) | $85.7 \pm 6.3$ | $67.9 \pm 8.8$ | $86.3 \pm 6.6^{ns}$ | $70.2 \pm 7.8^{**}$ |
| Pancreas | $85.3 \pm 7.3$ | $63.6 \pm 15.2$ | $85.5 \pm 6.6^{ns}$ | $64.9 \pm 12.5^{*}$ |
| Average | $90.3 \pm 6.4$ | $69.6 \pm 14.7$ | $90.4 \pm 8.2$ | $70.7 \pm 14.1$ |

| **TotalSegmentator** [89] | Trained on 16 partially labeled datasets | | Trained on AbdomenAtlas-8K | |
|---|---|---|---|---|
| | mDSC (%) | mNSD (%) | mDSC (%) | mNSD (%) |
| Spleen | $93.2 \pm 11.0$ | $59.4 \pm 11.7$ | $94.8 \pm 8.4^{****}$ | $65.0 \pm 12.3^{****}$ |
| Right Kidney | $91.2 \pm 10.5$ | $55.6 \pm 13.9$ | $91.8 \pm 9.8^{**}$ | $57.5 \pm 13.2^{***}$ |
| Left Kidney | $89.4 \pm 14.2$ | $58.7 \pm 13.6$ | $90.4 \pm 12.7^{***}$ | $59.9 \pm 13.3^{***}$ |
| Gall Bladder | $82.2 \pm 15.5$ | $45.2 \pm 11.9$ | $82.9 \pm 15.9^{*}$ | $48.3 \pm 12.6^{****}$ |
| Liver | $94.0 \pm 10.0$ | $53.1 \pm 10.5$ | $94.6 \pm 9.3^{ns}$ | $54.4 \pm 10.8^{*}$ |
| Stomach | $82.8 \pm 17.4$ | $43.6 \pm 11.9$ | $84.2 \pm 18.0^{**}$ | $46.8 \pm 12.5^{****}$ |
| Aorta | $72.1 \pm 19.9$ | $47.9 \pm 13.0$ | $75.7 \pm 19.5^{****}$ | $53.5 \pm 14.2^{****}$ |
| Postcava (IVC) | $73.7 \pm 18.9$ | $47.5 \pm 14.2$ | $74.7 \pm 18.7^{**}$ | $50.2 \pm 15.3^{****}$ |
| Pancreas | $80.6 \pm 15.4$ | $40.9 \pm 10.3$ | $82.4 \pm 15.7^{***}$ | $46.5 \pm 10.8^{****}$ |
| Average | $84.4 \pm 14.8$ | $50.2 \pm 12.3$ | $85.7 \pm 14.2$ | $53.6 \pm 12.8$ |

| **JHH** [92] | Trained on 16 partially labeled datasets | | Trained on AbdomenAtlas-8K | |
|---|---|---|---|---|
| | mDSC (%) | mNSD (%) | mDSC (%) | mNSD (%) |
| Spleen | $94.7 \pm 1.8$ | $72.0 \pm 8.5$ | $95.0 \pm 1.5^{*}$ | $74.4 \pm 8.4^{****}$ |
| Right Kidney | $92.2 \pm 5.2$ | $65.2 \pm 9.3$ | $92.2 \pm 5.0^{ns}$ | $64.7 \pm 8.9^{**}$ |
| Left Kidney | $91.7 \pm 1.8$ | $65.7 \pm 9.0$ | $91.6 \pm 1.8^{ns}$ | $65.1 \pm 9.3^{**}$ |
| Gall Bladder | $82.6 \pm 12.6$ | $60.8 \pm 15.4$ | $83.8 \pm 10.5^{**}$ | $60.1 \pm 15.8^{*}$ |
| Liver | $95.1 \pm 7.7$ | $66.0 \pm 8.8$ | $95.1 \pm 7.7^{ns}$ | $66.2 \pm 8.6^{ns}$ |
| Stomach | $91.6 \pm 6.5$ | $53.1 \pm 9.8$ | $92.4 \pm 6.2^{**}$ | $55.7 \pm 9.7^{****}$ |
| Aorta | $72.1 \pm 10.2$ | $55.5 \pm 10.1$ | $74.2 \pm 10.5^{****}$ | $57.5 \pm 10.3^{**}$ |
| Postcava (IVC) | $77.2 \pm 11.0$ | $54.9 \pm 9.5$ | $77.7 \pm 10.9^{*}$ | $57.5 \pm 10.3^{**}$ |
| Pancreas | $78.8 \pm 12.4$ | $52.3 \pm 8.1$ | $79.3 \pm 11.3^{***}$ | $55.2 \pm 8.8^{****}$ |
| Average | $86.2 \pm 7.7$ | $60.6 \pm 9.8$ | $86.8 \pm 7.3$ | $61.8 \pm 9.9$ |

$^{ns}$ $P > 0.05$    $^{*}$ $P \le 0.05$    $^{**}$ $P \le 0.01$    $^{***}$ $P \le 0.001$    $^{****}$ $P \le 0.0001$

Table 7: **Generalization of AI models trained on AbdomenAtlas-8K.** We directly applied AI trained on AbdomenAtlas-8K (public) to JHH (unseen, private) and compared its performance with AI trained on JHH.

| SwinUNETR [84] | Trained on JHH (private) | | Trained on AbdomenAtlas-8K (public) | |
| --- | --- | --- | --- | --- |
| | mDSC (%) | mNSD (%) | mDSC (%) | mNSD (%) |
| Spleen | $92.7 \pm 12.2$ | $76.7 \pm 16.6$ | $94.2 \pm 4.7$ | $73.1 \pm 10.6$ |
| Right Kidney | $92.6 \pm 10.4$ | $80.5 \pm 11.6$ | $90.8 \pm 11.6$ | $64.8 \pm 10.9$ |
| Left Kidney | $91.8 \pm 9.2$ | $78.8 \pm 11.4$ | $91.0 \pm 4.3$ | $65.3 \pm 9.7$ |
| Gall Bladder | $74.5 \pm 23.9$ | $54.1 \pm 24.0$ | $81.9 \pm 13.4$ | $57.2 \pm 17.5$ |
| Liver | $96.2 \pm 1.1$ | $69.5 \pm 8.6$ | $96.1 \pm 1.0$ | $65.2 \pm 8.0$ |
| Stomach | $91.7 \pm 7.8$ | $57.8 \pm 14.4$ | $93.1 \pm 2.6$ | $56.5 \pm 10.1$ |
| Aorta[†] | $87.6 \pm 7.6$ | $80.5 \pm 11.7$ | $73.1 \pm 12.0$ | $58.0 \pm 12.7$ |
| Postcava (IVC) | $70.4 \pm 18.0$ | $51.2 \pm 17.2$ | $77.9 \pm 10.3$ | $54.6 \pm 10.5$ |
| Pancreas | $65.8 \pm 23.2$ | $49.3 \pm 17.4$ | $81.1 \pm 7.7$ | $52.5 \pm 8.1$ |
| Average | $84.8 \pm 12.6$ | $66.5 \pm 14.8$ | $86.5 \pm 7.5$ | $60.8 \pm 10.9$ |

| UNETR [28] | Trained on JHH (private) | | Trained on AbdomenAtlas-8K (public) | |
| --- | --- | --- | --- | --- |
| | mDSC (%) | mNSD (%) | mDSC (%) | mNSD (%) |
| Spleen | $89.9 \pm 8.2$ | $62.3 \pm 18.6$ | $94.0 \pm 4.6$ | $72.2 \pm 11.1$ |
| Right Kidney | $79.3 \pm 15.2$ | $58.7 \pm 16.4$ | $92.0 \pm 4.2$ | $63.3 \pm 8.7$ |
| Left Kidney | $83.6 \pm 13.4$ | $63.2 \pm 14.8$ | $91.5 \pm 2.2$ | $64.2 \pm 9.6$ |
| Gall Bladder | $72.4 \pm 17.2$ | $50.3 \pm 15.8$ | $81.3 \pm 12.1$ | $55.9 \pm 16.7$ |
| Liver | $93.4 \pm 3.7$ | $55.7 \pm 12.8$ | $96.0 \pm 0.9$ | $64.2 \pm 7.8$ |
| Stomach | $84.9 \pm 10.0$ | $38.8 \pm 9.8$ | $92.9 \pm 2.6$ | $53.1 \pm 9.5$ |
| Aorta[†] | $83.9 \pm 9.3$ | $71.6 \pm 14.1$ | $74.0 \pm 12.6$ | $57.8 \pm 12.8$ |
| Postcava (IVC) | $58.3 \pm 20.1$ | $41.2 \pm 13.0$ | $76.2 \pm 11.1$ | $52.3 \pm 10.1$ |
| Pancreas | $62.1 \pm 19.7$ | $43.2 \pm 11.6$ | $81.2 \pm 7.9$ | $52.6 \pm 7.5$ |
| Average | $78.6 \pm 13.0$ | $53.9 \pm 14.1$ | $86.6 \pm 6.5$ | $59.5 \pm 10.4$ |

| U-Net [73] | Trained on JHH (private) | | Trained on AbdomenAtlas-8K (public) | |
| --- | --- | --- | --- | --- |
| | mDSC (%) | mNSD (%) | mDSC (%) | mNSD (%) |
| Spleen | $90.0 \pm 12.5$ | $72.0 \pm 17.9$ | $94.7 \pm 2.8$ | $74.0 \pm 9.8$ |
| Right Kidney | $92.3 \pm 7.3$ | $79.4 \pm 11.2$ | $91.8 \pm 6.2$ | $63.5 \pm 9.2$ |
| Left Kidney | $89.7 \pm 10.4$ | $74.4 \pm 14.1$ | $91.5 \pm 2.6$ | $65.7 \pm 9.5$ |
| Gall Bladder | $76.3 \pm 17.7$ | $56.0 \pm 20.0$ | $82.8 \pm 11.2$ | $58.5 \pm 14.7$ |
| Liver | $95.1 \pm 2.4$ | $63.7 \pm 11.5$ | $96.1 \pm 0.9$ | $64.2 \pm 7.9$ |
| Stomach | $90.8 \pm 6.3$ | $53.1 \pm 13.4$ | $93.2 \pm 2.9$ | $56.0 \pm 9.8$ |
| Aorta[†] | $84.6 \pm 12.4$ | $76.2 \pm 14.7$ | $74.3 \pm 11.1$ | $57.4 \pm 12.4$ |
| Postcava (IVC) | $73.0 \pm 14.3$ | $50.5 \pm 17.4$ | $78.2 \pm 10.1$ | $55.1 \pm 10.5$ |
| Pancreas | $71.2 \pm 17.6$ | $52.6 \pm 12.8$ | $82.7 \pm 6.0$ | $54.2 \pm 8.0$ |
| Average | $84.8 \pm 11.2$ | $64.2 \pm 14.8$ | $87.3 \pm 6.0$ | $61.0 \pm 10.2$ |

| SegResNet [12] | Trained on JHH (private) | | Trained on AbdomenAtlas-8K (public) | |
| --- | --- | --- | --- | --- |
| | mDSC (%) | mNSD (%) | mDSC (%) | mNSD (%) |
| Spleen | $95.2 \pm 2.3$ | $78.8 \pm 9.5$ | $94.6 \pm 2.8$ | $74.1 \pm 9.8$ |
| Right Kidney | $93.9 \pm 6.5$ | $80.3 \pm 8.7$ | $92.0 \pm 5.9$ | $65.5 \pm 8.0$ |
| Left Kidney | $92.7 \pm 5.8$ | $78.2 \pm 8.7$ | $91.8 \pm 1.6$ | $65.7 \pm 7.9$ |
| Gall Bladder | $80.4 \pm 13.0$ | $55.7 \pm 16.4$ | $82.8 \pm 11.0$ | $58.2 \pm 16.3$ |
| Liver | $90.2 \pm 2.9$ | $58.1 \pm 7.5$ | $95.8 \pm 1.4$ | $63.4 \pm 8.3$ |
| Stomach | $92.8 \pm 5.1$ | $55.2 \pm 12.3$ | $93.3 \pm 2.9$ | $55.6 \pm 9.4$ |
| Aorta[†] | $85.4 \pm 6.9$ | $75.2 \pm 11.5$ | $73.9 \pm 12.2$ | $55.5 \pm 13.1$ |
| Postcava (IVC) | $75.4 \pm 9.8$ | $45.5 \pm 13.4$ | $77.2 \pm 10.6$ | $53.2 \pm 10.2$ |
| Pancreas | $82.6 \pm 6.8$ | $56.9 \pm 9.7$ | $81.5 \pm 6.5$ | $52.6 \pm 7.8$ |
| Average | $87.6 \pm 6.6$ | $64.9 \pm 10.9$ | $87.0 \pm 6.1$ | $60.4 \pm 10.1$ |

[†] During the training process using AbdomenAtlas-8K, the ground truth of **Aorta** exhibits variation across different public datasets, resulting in the AI's inability to learn as anticipated. Conversely, when training on the JHH dataset, the annotation protocol of **Aorta** remains relatively consistent.

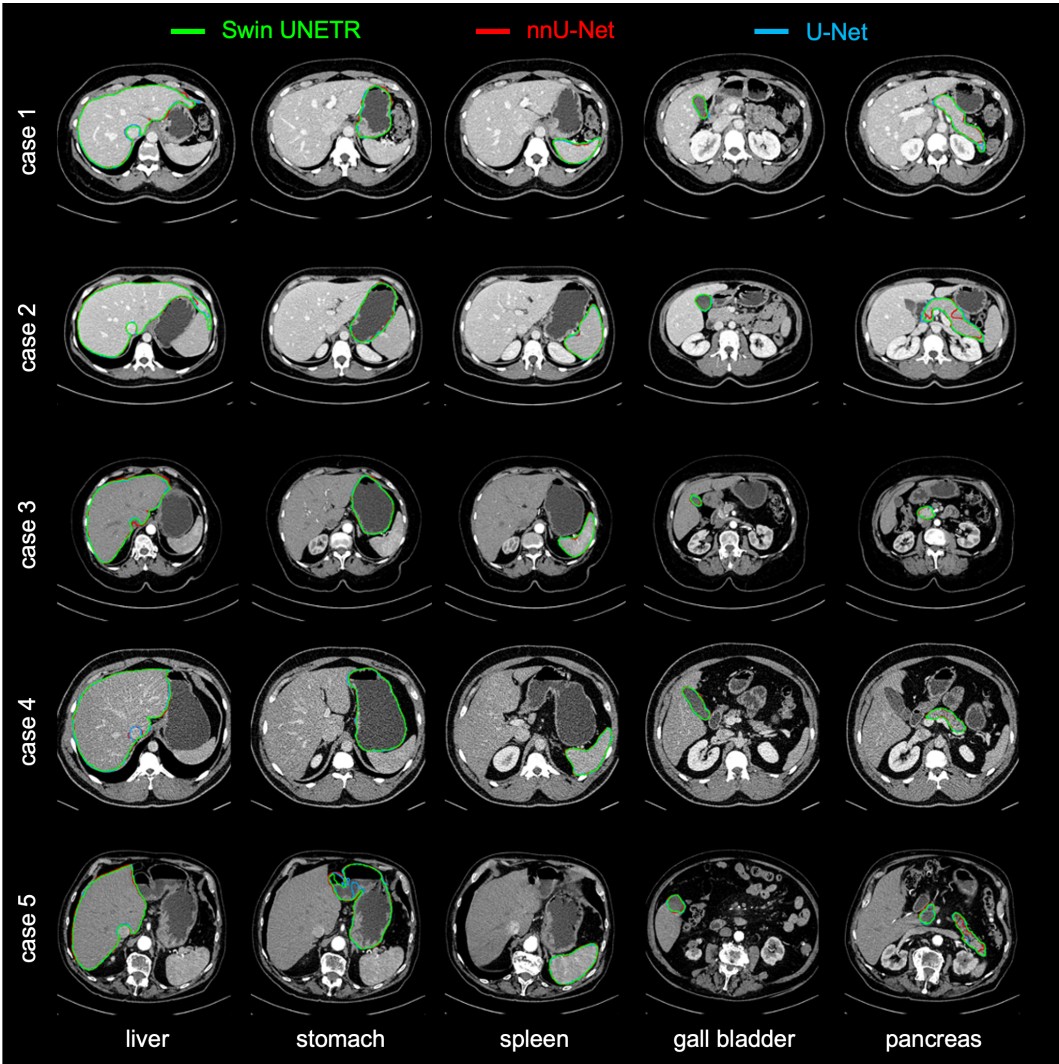

Figure 10: [Better viewed in color and zoomed in for details] **Variance in model predictions.** Our AbdomenAtlas-8K comprises an ensemble of three distinct segmentation architectures, thereby obviating the potential influence of architectural bias. As such, our AbdomenAtlas-8K stands out for its robustness and objectivity.

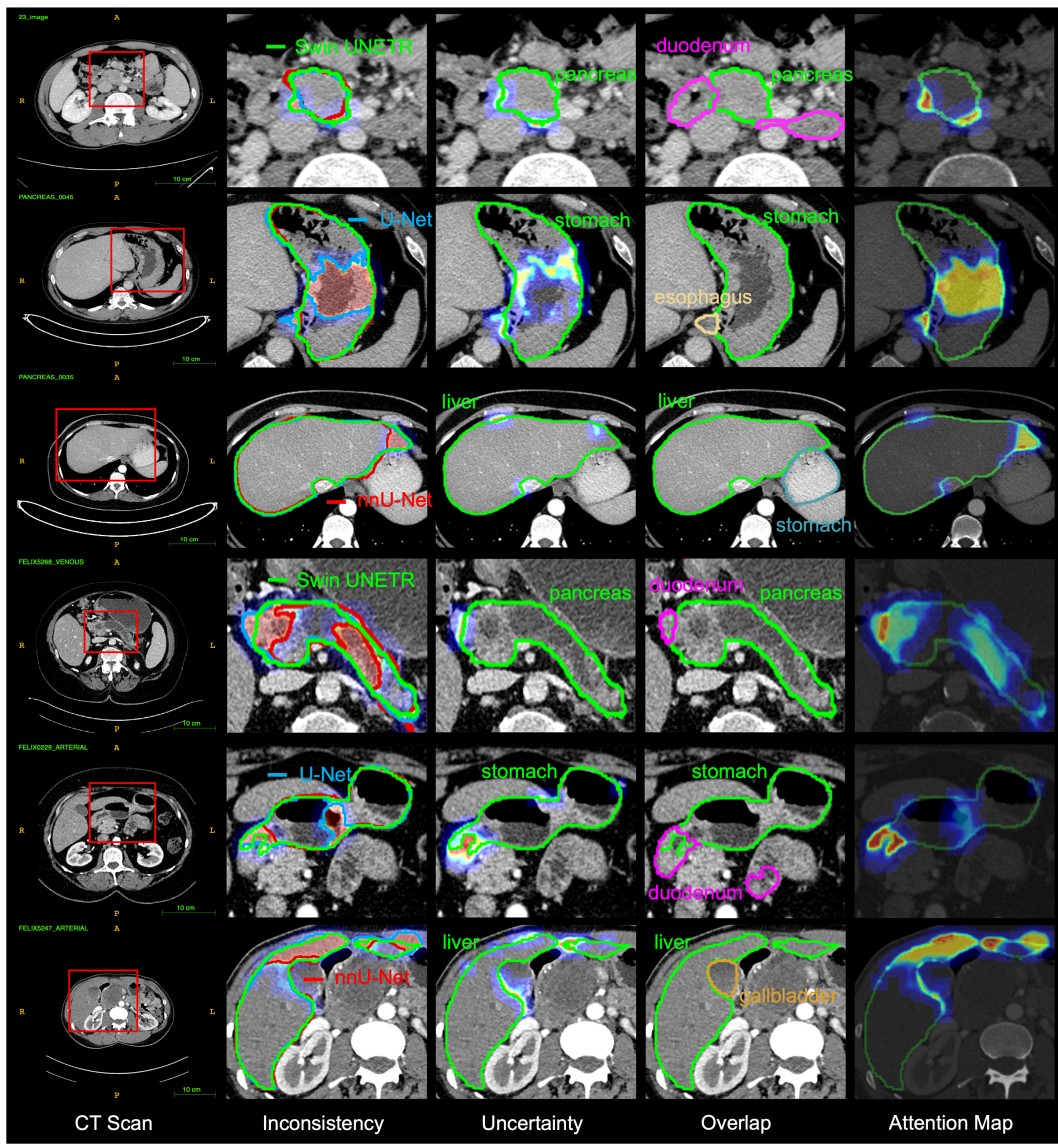

Figure 11: **Examples of attention map generation.** The attention map helps our annotator quickly and effectively locate regions with a high risk of prediction errors.

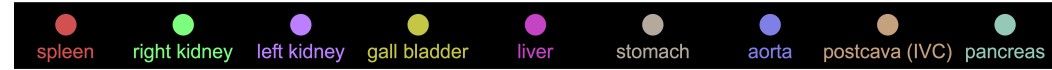

Figure 12: [Watch the animations by clicking them, better viewed with Acrobat/Foxit Reader] **Example annotated slices from AbdomenAtlas-8K**. AbdomenAtlas-8K includes annotations for eight organs, treating the left and right kidneys as a single organ.