# OpenReview forum: "AbdomenAtlas-8K: Annotating 8,000 CT Volumes for Multi-Organ Segmentation in Three Weeks"
_NeurIPS.cc/2023/Track/Datasets_and_Benchmarks — NeurIPS 2023 Datasets and Benchmarks Poster_

### Official Review · Reviewer_R6jP · 2023-07-04
**Review of "Annotating 8,000 Abdominal CT Volumes for Multi-Organ Segmentation in Three Weeks"**

**Rating:** 7
**Confidence:** 3
**Clarity:** The paper is clearly written.

**Strengths:**

The proposed algorithm is shown to provide rapid annotation possibilities.

The accuracy of the AI approach is compared favourably to annotations by two medical professionals.

The code is made available.

**Additional Feedback:**

Overall an interesting and potentially impactful paper.

**Correctness:**

It is concerning that the inter-annotator agreement between the two human annotators making changes to the AI generated annotations is rather low for some organs. This implies that two medical experts starting from the same initial AI annotation end up with quite different annotations for some organs.

This should be investigated and discussed in more detail. Why is it so much worse for some organs? Could this mean that the human annotators are being biased by the initial AI annotation?

**Documentation:**

The software is sufficiently described on the GitHub page. The dataset has not been released yet.

**Ethics:**

It is stated on the checklist that no human subjects were involved, which is not completely correct because at least two annotators were involved in the study.

**Limitations:**

A discussion on limitations is provided in the Discussion section.

The generalisability of the approach to other organs and imaging modalities is only mentioned briefly at the end of the conclusion. Some further discussion on the feasibility of this and the work that would be involved to adapt the software would be useful.

**Opportunities For Improvement:**

The dataset of 8000 annotated CT images created using the annotation tool is not available yet.

"Pseudo labels" should be defined.

The JHH dataset is referred to without defiining what it is. Is it a proprietary dataset?



**Relation To Prior Work:**

Extensive references are provided. The related work section discusses how this work differs from previous work.

**Summary And Contributions:**

The paper presents a new approach to speed up annotation of multiple organs in abdominal CT images by using multiple AI models to generate an initial segmentation and then only requiring the annotator to correct the annotation.

The contributions are;
- The new annotation algorithm based on active learning and three AI segmentation algorithms, which is claimed to reduce manual annotation effort and thereby speed up manual annotation by a factor of 533
- The source code is made available
- A taxonomy of common errors made by AI algorithms and annotators
- A dataset of 8000 manually annotated abdominal CT images (although this has not yet been released, it is on the ToDo list on the GitHub page for the code)
- An evaluation of the accuracy of the AI approach compared to manual annotation by two medical professionals

---

> ### Author Response · Authors · 2023-08-21
> **Responses to Reviewer R6jP**
>
> Many thanks for your thoughtful feedback and accolades on our paper: “overall an interesting and potentially impactful paper.”
>
> Q1: *The dataset of 8000 annotated CT images created using the annotation tool is not available yet.*
>
> A1: We used commercial software called [Pair](https://aipair.com.cn/) to perform annotation reviews and revisions, as specified in footnote #4. Acquiring this software requires purchasing a license for each individual user. Subsequently, we discovered that [MONAI-LABEL](https://monai.io/label.html) is also a very useful tool for the annotation task.
>
> Q2: *"Pseudo labels" should be defined.*
>
> A2: Thanks for your suggestion. Pseudo labels refer to organ labels predicted by AI models without any additional revision or validation by human annotators. We have now clarified this term in the revised paper.
>
> Q3: *The JHH dataset is referred to without defining what it is.*
>
> A3: The JHH dataset is a proprietary, multi-resolution, multi-phase collected from Johns Hopkins Hospital. In the updated paper, we have provided a more comprehensive clarification of the JHH dataset.
>
> Q4: *The generalisability of the approach to other organs and imaging modalities is only mentioned briefly at the end of the conclusion. Some further discussion on the feasibility of this and the work that would be involved to adapt the software would be useful.*
>
> A4: Thank you for initiating this discussion. We have undertaken several explorations to assess the generalisability of our approach, as outlined below.
>
> 1. **Generalisability to other organs/tumors.** After the paper was submitted, we continued to annotate 13 additional organs and vessels. (e.g., esophagus, portal vein and splenic vein, adrenal gland, duodenum, hepatic vessel, lung, colon, intestine, rectum, bladder, prostate, head of femur, and celiac trunk.), and different types of tumors (e.g., kidney tumor, liver tumor, pancreas tumor, hepatic vessel tumor, lung tumor, colon tumor, and kidney cyst). Leveraging our proposed approach can significantly accelerate the annotation of various anatomical structures. However, annotating more types of tumors remains challenging if solely using our approach; even human experts may face uncertainty in tumor annotation, particularly in the early stages of the disease. Consequently, further research is warranted. This could entail incorporating additional data sources such as radiology reports, biopsy results, and patient demographic information into the tumor annotation.
>
> 2. **Generalisability to other imaging modalities.** In this work, we have applied our approach to a number of CT volumes across different phases, including portal (44%), arterial (37%), pre- (16%), and post-contrast (3%) phases. Given the inherent differences in these scans, we believe that our active learning procedure can be extended to datasets from other imaging modalities, including MRI, Ultrasound, Histopathology images, etc.
>
> 3. **Adapt to the software.** Our active learning procedure is being integrated into open-source software such as [MONAI-LABEL](https://monai.io/label.html) at NVIDIA and [ChimeraX](https://www.cgl.ucsf.edu/chimerax/) at UCB/UCSF.
>
> Q5: *Why is the inter-annotator agreement so much worse for some organs? Could this mean that the human annotators are being biased by the initial AI annotation?*
>
> A5: Figure 8(b) presented the inter-annotator agreement across the two human experts and one AI model. Overall, the agreement between the two human experts is fairly high (DSC > 0.9). Several organs have a relatively lower agreement, such as the gallbladder (DSC = 0.85) and pancreas (DSC = 0.81). It is because the boundaries of these two organs are often blurry and ambiguous.
>
> The purpose of investigating the inter-annotator agreement in Figure 8(b) is to assess the quality of AI predictions for organ segmentation. Therefore, in Figure 8(b), we asked the two radiologists to annotate these organs from scratch without the use of initial AI annotation—**the AI annotation will not bias the radiologists**. As a result, for each organ, we obtained two annotations made independently by the two human experts and one annotation predicted by the AI model. Figure 8(b) presented the mutual DSC score of each pair, showing that AI can segment these organs with a similar variance to human experts.
>
> Q6: *The dataset has not been released yet.*
>
> A6: The dataset will be publicly available once the paper is accepted. All requisite code for AbdomenAtlas-8K creation has been released on GitHub, and we have confirmed with lawyers at Johns Hopkins University for the permissions necessary for distributing the full annotations. To clarify, we will only disseminate the annotations of the CT volumes separately, and users will retrieve the original CT volumes, if needed, from the original sources (websites). Everything we intend to create and license out will be in separate files, and no modifications are necessary to the original CT volumes.

---

### Official Review · Reviewer_nhxF · 2023-07-17
**A comprehensive benchmarking dataset for multi-organ segmentation**

**Rating:** 9
**Confidence:** 4
**Clarity:** Yes

**Strengths:**

The construction of the large dataset and the use of active learning for annotation are highlights of the study.

**Additional Feedback:**

None

**Correctness:**

I believe the dataset is constructed in a sound way and the evaluation methods were largely appropriately designed.

**Documentation:**

Code for testing with own dataset is available, however, I did not find the benchmarking dataset itself in the github repository.

**Limitations:**

No obvious limitations were identified in the body of work performed. The potential of labeling tumor in tissues in the future to reduce false positive labels, has already been discussed in the manuscript.

**Opportunities For Improvement:**

The quality of the dataset is impressive and the construction of the ground truth is convincing. I noticed that in Figure 5, the attention map seems to contain regions that do not belong to FP/FN labels or cases of ambiguity. If such regions were also subject to human curation, it may involve unnecessary additional work.

**Relation To Prior Work:**

Previous datasets were sufficiently cited and compared to the current dataset, which demonstrated the comprehensiveness of the current dataset construction.

**Summary And Contributions:**

In this paper, Qu et al proposed a benchmark dataset of over 8000 volumes of abdominal CT images as well as a computer-assisted multi-organ segmentation result. With the help of attention guidance map based on the consensus of three SI algorithms, human annotators can generate ground truth segmentation significantly faster than complete manual annotation. The average segmentation result generated by AI algorithms with iterative input from human annotators further provide higher consistency of segmentation result compared to individual annotators. As currently the largest dataset, the work significantly contributed to the field of abdominal CT imaging and can serve as the basis of the development of effective AI algorithm in the future.

---

> ### Author Response · Authors · 2023-08-21
> **Responses to Reviewer nhxF**
>
> We are grateful for your **“strong accept”** and accolades, especially “no obvious limitations were identified…”, “...dataset is constructed in a sound way”, “...evaluation methods were largely appropriately designed”, and “previous datasets were sufficiently cited and compared…the comprehensiveness of the current dataset construction”. Below is our response to your two discussions.
>
> Q1: *I noticed that in Figure 5, the attention map seems to contain regions that do not belong to FP/FN labels or cases of ambiguity. If such regions were also subject to human curation, it may involve unnecessary additional work.*
>
> A1: We greatly appreciate your feedback and providing this concern. Reducing such regions is one of the main focuses in the active learning literature. Compared with random selection and revision, our proposed method has yielded a significant reduction in human curation. But certainly, there are regions where AI made correct predictions but our method still asked humans to review and where AI made mistakes but our method cannot detect them. To analyze and overcome this problem, we took two actions in this study.
>
> 1. We have quantified these regions in Table 1, suggesting that for some organs, our method presented high sensitivity and precision (both > 0.9), while some organs had relatively lower sensitivity (e.g., liver) and precision (e.g., spleen, gallbladder, and IVC). Consequently, the active learning methods for these organs require further investigation.
>
> 2.  Two junior radiologists (3-year experience) took the responsibility to look through the entire AbdomenAtlas-8K once the active learning procedure is completed. The radiologists will make a revision if the labels were incorrect (i.e., our method missed the regions), but such revisions were seldom required, with only 55 out of 8,448 instances needing adjustments.
>
> Q2: *Code for testing with own dataset is available, however, I did not find the benchmarking dataset itself in the github repository.*
>
> The dataset will be made publicly available once the paper has been accepted. At the moment, all requisite code for AbdomenAtlas-8K creation has been released on GitHub, and we have confirmed with lawyers at Johns Hopkins University for the permissions necessary for distributing the full annotations. To clarify, we will only disseminate the annotations of the CT volumes separately, and users will retrieve the original CT volumes, if needed, from the original sources (websites). Everything we intend to create and license out will be in separate files, and no modifications are necessary to the original CT volumes.

---

### Official Review · Reviewer_pW4Y · 2023-07-20
**Review for #288**

**Rating:** 7
**Confidence:** 5
**Clarity:** The paper is overall very clear.

**Strengths:**

-The strategy of combining complementary strengths of multiple segmentation models trained on the same limited dataset is compelling.
-The proposed attention map for interactive segmentation refinement of human raters is useful and substantially reduces annotation time compared to a complete manual annotation.
-The visualisations are thoroughout very well done and informative.

**Additional Feedback:**

The authors performed a very thorough rebuttal and improved the paper quality during this revision. I have therefore improved my rating from 6 to 7.

**Correctness:**

-This would also fall under “limitations”: The process is stopped after the highest priority scan is deemed “error free” by the expert. But the priority list is computed by the algorithm itself, so there’s no guarantee that the remaining 92.5% are correct. I could think of multiple failure cases, where the proposed attention map score is “incorrectly” predicting no error, e.g. because the image is seemingly easy for all three DL models but contains an important abnormality.
-As mentioned above, the main claim of the paper: that a larger CT dataset is necessary is unfortunately not convincingly demonstrated. In Appendix A the average Dice of training on the previously available datasets vs the new Abdomen8K is equivalent 90.3% vs 90.4% and no statistical test shown. In addition the new dataset reduces the number of labels compared to TotalSegmentator (Abominal 24➞8)

**Documentation:**

As mentioned it is unclear whether the JHH dataset will be made available under CC licence. And there are still quite a few todos in the github repository. The licence of the 5k internal JHU scans is given as “15. JHH (private) [82]”, but the github claims the whole dataset will be released.

**Ethics:**

It did not find any information on ethical approval for releasing the new JHU dataset.

**Limitations:**

-The reduction of abdominal labels to 8 is a limitation. Many smaller anatomical structures (adrenal glands, etc) are also of clinical interest and in comparison to the TotalSegmentator which provides 104 anatomies this is actually a step backwards.
-It remains somewhat unclear whether all the data will be publicly available upon acceptance. Since for the largest subset of the Abdomen8k JHH is is noted that no licence is available.
-An important aspect of the active labelling procedure (5) is not described, the sentence is incomplete.
-The comparison of training with this new dataset compared to previous partially labelled ones in Appendix A does not show a higher Dice 90.3 vs 90.4% and no statistical tests demonstrate the usefulness of this expanded dataset.

**Opportunities For Improvement:**

-It is a bit odd that the authors seem to criticise the TotalSegmentator in the begging for using a single-model automatic segmentation for finally annotating the majority of scans in the dataset, since they actually do the same (just using an ensemble of three models) and only let annotators correct 400/8000 volumes.

**Relation To Prior Work:**

The relations and differences to the TotalSegmentator could be further discussed. Is it really the fact that employing only an nnUNet model leads to high correlation or would three nnUNets trained with different initialisations (ensemble) also work?

**Summary And Contributions:**

The submission provides an interesting concept to efficiently annotate a very large CT dataset.

---

> ### Author Response · Authors · 2023-08-21
> **Responses to Reviewer pW4Y (1/3)**
>
> We are grateful for recognizing the strengths of our paper, “the strategy…is compelling,” “the proposed attention map…is useful and substantially reduces annotation time,” and “the visualisations are thoroughout very well done and informative.” We have fixed all the issues that you kindly pointed out.
>
> Q1: *It is a bit odd that the authors seem to criticise the TotalSegmentator in the begining for using a single-model automatic segmentation for finally annotating the majority of scans in the dataset, since they actually do the same (just using an ensemble of three models) and only let annotators correct 400/8000 volumes.*
>
> A1: In TotalSegmentator, the assertion, *“all 1,204 CT examinations had annotations that were manually reviewed and corrected whenever necessary”* (quote from the original paper) stated two crucial points. Therefore, (1) the labels were largely generated by a **single** nnU-Net re-trained continually and (2) the annotators must review and revise **every** CT volume whenever necessary. Thereby, the main difference between TotalSegmentator and AbdomenAtlas-8K is two-fold:
>
> 1. **Reducing architectural biases.** The labels in AbdomenAtlas-8K were averaged from **three** segmentation architectures. Depending solely on nnU-Net could introduce a potential label bias favoring the nnU-Net architecture. This means that whenever TotalSegmentator is employed for benchmarking, nnU-Net would always outperform other segmentation architectures (e.g., UNETR, TransUNet, SwinUNet, etc.). This observation has also been made in several publications that used the TotalSegmentator dataset. For example, Table 3 in [[Huang et al., 2023](https://arxiv.org/abs/2304.06716)] showed that nnFormer, UNETR, and SwinUNETR were all outperformed by nnU-Net in TotalSegmentator (of course their proposed architecture surpassed nnU-Net, but this required independent validation). Hence, the use of an ensemble of three (or even more) models was crucial to reduce such an architectural bias.
>
> 2. **Reducing annotation time.** As outlined in the TotalSegmentator paper, the annotators must review and revise (when necessary) every CT volume (1204/1204 volumes) without priority. In contrast, we have introduced an innovative active learning procedure that identifies the most crucial CT volumes and significant regions within them, guiding annotators to put their efforts more accurately. Consequently, annotators are now only tasked with revising **400/8448** volumes. This represents a substantial reduction (from 30.8 years to three weeks) in the radiologists' workload through algorithmic innovation.
>
> Q2: *The reduction of abdominal labels to 8 is a limitation. Many smaller anatomical structures (adrenal glands, etc) are also of clinical interest and in comparison to the TotalSegmentator which provides 104 anatomies this is actually a step backwards.*
>
> A2: Thank you for your comment. We respectfully disagree that this is a step backward. This paper advanced an active learning procedure that enabled us to produce large-scale, fully-annotated datasets in a short span of time. AbdomenAtlas-8K was one of the demonstrations of the practical use of the procedure. In fact, our proposed active learning procedure can be adapted to create annotations for a broader range of anatomical structures (including smaller structures) and even cancerous tumors. In contrast, while TotalSegmentator is a large dataset with numerous annotated structures, extending its annotation methodology to various other tasks can be time-consuming and therefore not as straightforward. Moreover, AbdomenAtlas-8K itself (as a dataset contribution) showed unique advantages in comparison with TotalSegmentator.
>
> 1. **Significant larger number of CT volumes**. Compared with 1204 CT volumes in TotalSegmentator, AbdomenAtlas-8K contains 8448 CT volumes, which is about eight times larger than TotalSegmentator.
>
> 2. **Much more diverse population**. TotalSegmentator collected CT volume in University Hospital Basel (biased on the Central European population), while AbdomenAtlas-8K unified medical datasets from more than 26 hospitals worldwide. The higher diversity in AbdomenAtlas-8K significantly prevents the dataset bias on certain populations.

---

> ### Author Response · Authors · 2023-08-21
> **Responses to Reviewer pW4Y (2/3)**
>
> Q3: *It remains somewhat unclear whether all the data will be publicly available upon acceptance. Since for the largest subset of the Abdomen8k JHH is noted that no license is available.*
>
> A3: We commit to releasing 3,410 of the 8,448 CT volumes upon acceptance. The dataset usage and license are detailed in Table 3 (supplementary). While the JHH dataset cannot be released due to numerous regulation constraints, we continued to apply our proposed active learning procedure to newly released datasets after we submitted this paper. More specifically, we have fully annotated the FLARE’23 (4,000 CT volumes), AATTCT-IDS (300 CT volumes), CTSpine1K (1,005 CT volumes), and Abdominal Trauma Detection (3,147 CT volumes) datasets by integrating the best of our AI algorithms and radiologist revision. Again, our main contribution is an efficient active learning procedure that can be used to quickly create large-scale annotated datasets as needed. Looking forward, we anticipate a significant increase in the number of publicly available abdominal CT volumes (fully labeled, partially labeled, or unlabeled). Over time, our AbdomenAtlas-8K dataset will be much larger than 8,448 CT volumes and go beyond eight types of annotated organs.
>
> Q4: *An important aspect of the active labelling procedure (5) is not described, the sentence is incomplete.*
>
> A4: Thank you for detecting this issue. We have revised the sentence: “The annotator revises pseudo labels of eight target organs based on the attention maps and prioritized list.”
>
> Q5: *The comparison of training with this new dataset compared to previous partially labelled ones in Appendix A does not show a higher Dice 90.3 vs 90.4% and no statistical tests demonstrate the usefulness of this expanded dataset.*
>
> A5: The marginal improvement in the FLARE’23 dataset can be due to many reasons. For example, the performance of the eight specific organs is already very high (DSC > 90%) and training with more annotations may not yield significant returns.
>
> Moreover, we also recognized that limiting our evaluation exclusively to the FLARE'23 dataset may not provide a comprehensive assessment, as its small sample size (*N* = 300) may not present the full spectrum of different domains. To improve, we have evaluated the trained models on two additional (unseen) datasets with larger sample sizes (*N*): TotalSegmentator (*N* = 1,204) and JHH (*N* = 5,281). Following your suggestion, we have conducted a statistical analysis of the comparison. The mean, standard deviation, and *p*-value are reported in the revised Table 6. We obtained a more noticeable benefit from a larger scale evaluation when training AI models on AbdomenAtlas-8K over the previously partially labeled ones.
>
> Q6: *But the priority list is computed by the algorithm itself, so there’s no guarantee that the remaining 92.5% are correct. I could think of multiple failure cases, where the proposed attention map score is “incorrectly” predicting no error, e.g. because the image is seemingly easy for all three DL models but contains an important abnormality.*
>
> A6: Thank you for providing this insightful comment. It is true that the remaining 92.5% have a potentially important abnormality in all three models. Our solutions are in two dimensions:
>
> 1. **Quickly review the entire dataset**. Two junior radiologists (3-year experience) were responsible for looking through the entire AbdomenAtlas-8K once the active learning procedure was completed. The radiologists would make a revision if the labels were incorrect, but such revisions were seldom required, with only 55 out of 8,448 instances needing adjustments. This strategy guaranteed the automated AI annotation quality in the remaining 92.5%.
>
> 2. **Enrich the AI architectures used for dataset construction**. We plan to unify segmentations predicted by more AI architectures. This strategy can significantly attenuate the prediction errors made by specific AI architectures in the remaining 92.5%.

---

> ### Author Response · Authors · 2023-08-21
> **Responses to Reviewer pW4Y (3/3)**
>
> Q7: *The relations and differences to the TotalSegmentator could be further discussed. Is it really the fact that employing only an nnUNet model leads to high correlation or would three nnUNets trained with different initialisations (ensemble) also work.*
>
> A7: Thank you for your suggestions, we have now made an explicit comparison between AbdomenAtlas-8K and TotalSegmentator in the revised related work section.
>
> 1. **Relations and differences to the TotalSegmentator.** Both TotalSegmentator and AbdomenAtlas-8K involved human annotators manually revising AI predictions. The difference mainly lay in the active learning procedure. AbdomenAtlas-8K provided human annotators with a prioritized list and attention maps, suggesting the most important CT volumes to be revised and the saliency region that needs to be reviewed.
>
> 2. **Is it really the fact that employing only a nnUNet leads to a high correlation?** Yes, whenever TotalSegmentator is used for benchmarking, we found that nnU-Net would consistently outperform other segmentation architectures (e.g., UNETR, TransUNet, SwinUNet, etc.). This was not often observed in other datasets. We have discussed this point in A1.
>
> 3. **Would three nnU-Nets also work?** Yes, we believe that three nnU-Nets trained with different dataset ensembles could potentially solve the label bias issue. However, this approach might not guarantee that each nnU-Net model receives datasets containing the organ labels of our interest. We chose to use three distinct AI architectures and train on all 14 public partially-labeled medical datasets, which not only solves the label bias problem but also encompasses all the organs we are concerned about.
>
> Q8: *As mentioned it is unclear whether the JHH dataset will be made available under CC licence. And there are still quite a few todos in the github repository. The licence of the 5k internal JHU scans is given as “15. JHH (private) [82]”, but the github claims the whole dataset will be released.*
>
> As clarified in Q3, we commit to releasing 3,410 of the 8,448 CT volumes upon acceptance. We have revised our GitHub and manuscript to make this clearer. As of now, we have fully released the code and trained models that can be used to produce the dataset using the proposed active learning procedure. Of course, the main process is radiologist-in-the-loop revision, which requires extensive efforts (recruiting radiologists and human revision) in addition to what we already have on GitHub.
>
> **Reference**
>
> [1] Huang, Z., Wang, H., Deng, Z., Ye, J., Su, Y., Sun, H., He, J., Gu, Y., Gu, L., Zhang, S. and Qiao, Y., STU-Net: Scalable and Transferable Medical Image Segmentation Models Empowered by Large-Scale Supervised Pre-training. arXiv preprint arXiv. 2023

---

> > ### Comment · Reviewer_pW4Y · 2023-08-27
> > **Answer to responses for pW4Y**
> >
> > I highly appreciate the detailed response and clarifications provided by the authors. I am now even more convinced that the dataset will provide a meaningful addition to the field of medical image segmentation. Nevertheless, I feel that the authors have made some contradictory arguments in their rebuttal and would encourage them to see their Abdomen-8K as complementary to the TotalSegmentator dataset rather than trying to argue it is better in every aspect. To be specific: as outlined in answer 1/3 (Reducing annotation time) the TotalSegmentator is criticised for excessive checking of label quality: "must review and revise (when necessary) every CT volume" in answer 2/3 the authors state "Two junior radiologists (3-year experience) were responsible for looking through the entire AbdomenAtlas-8K" hence they used the same approach.
> >
> > I am also not convinced by answer Q2 in 1/3: I agree that the "paper advanced an active learning procedure" and the greater variability of populations by pooling data from different hospitals is a valuable contribution. But I would also hope that the authors could acknowledge that reducing the number of anatomies from 104 to 8 is a disadvantage. Checking and active corrections would indeed take longer with more structures but the benefits due to better coverage and inclusion of more bones are significant.
> >
> > The detailed answer about ensembles is helpful - the additional results in Table 6 (validation on TotalSegmentator) are convincing.
> >
> > I'm inclined to raise my score if the authors could somehow alleviate the remaining concerns about pros and cons of Abdomen-8K vs TotalSegmentator

---

> ### Author Response · Authors · 2023-08-28
> **Responses to Reviewer pW4Y (second round)**
>
> We will attach our second-round revised manuscript on August 29. Below are our responses to your concerns.
>
> Q1: *Concerns about pros and cons of Abdomen-8K vs. TotalSegmentator.*
>
> A1: We appreciate your perspective, and upon reflection, concur with the contradiction pointed out. Our intention was not to undermine the significance of TotalSegmentator, but rather to highlight the distinct features of AbdomenAtlas-8K. We regret any impression of being overly defensive. In response, we have revised our related work once again to make it clearer. A balanced literature review honors a great tribute to the scientific traditions that we all hold dearly.
>
> 1. **Number of annotated classes.** Both datasets serve pivotal roles in medical image segmentation. The TotalSegmentator's capacity to generate pseudo labels for an impressive 104 classes is the foundational groundwork necessary for initializing our active learning process. We acknowledge that our current focus on eight organs in AbdomenAtlas-8K should be considered as a limitation, particularly for those wishing to employ our active learning procedure across a broader spectrum of structures. Your feedback is invaluable.
>
> 2. **Reviewing the label quality.** It is true that both AbdomenAtlas-8K and TotalSegmentator require professionals to review the labels of the entire dataset for quality control—a very time-consuming undertaking indeed—especially for over 8,000 CT volumes and the sheer diversity of classes. While the time invested in *reviewing* appears equivalent between AbdomenAtlas-8K and TotalSegmentator, our active learning procedure reduces the time allocated for manual *revising*. Such revisions were seldom required during the review phase, with only **55 out of 8,448 (0.65%)** CT volumes needing adjustments from the two junior radiologists.
>
> Concurrently, we are actively expanding the classes covered by the AbdomenAtlas-8K. Starting with the set of 104 classes found in TotalSegmentator, we aim to further diversify the range of classes covered. We believe this progressive enhancement will further bridge the gap and accentuate the complementary nature of our dataset to TotalSegmentator and other public datasets.

---

> > ### Comment · Reviewer_pW4Y · 2023-08-29
> > **Upgraded score**
> >
> > I would like to thank the authors for further improving on the discussion about pros and cons for Abdomen-8K and TotalSegmentator and have upgraded my score. I now feel confident that the value as well as potential limitations of the newly proposed dataset as well as the benefits of the active labelling methods have become clearer.

---

### Official Review · Reviewer_Tr7Z · 2023-07-21
**Issues with Contribution and What Constitutes the Released Dataset**

**Rating:** 7
**Confidence:** 4

**Strengths:**

The paper is well written and easy to follow. Method is used to create a large-scale segmentation dataset of eight organs over a large number of CT scans. Since annotating medical images and release of data is typically a large burden which prevents AI model development, such a dataset will be very useful to facilitate development and training of AI for medical applications.

**Additional Feedback:**

None.

**Clarity:**

The paper is well-written and the documentation is through. Annotation code is released on a public Github page. The data itself was not released, but the authors promised to release it son.
As discussed above (see Opportunities for Improvement), information of the data reuse from previous datasets need to be clarified.

**Correctness:**

As discussed above (see Opportunities for Improvement), information of the data reuse from previous datasets need to be clarified. It does not appear that this is a novel dataset, rather, a release of improved data labels on existing datasets.
It was not immediately clear to me that the proposed labels offered more accurate information over existing labels (see Opportunities for Improvement), and I would encourage the authors to address this point in their rebuttal.

**Documentation:**

The data is not released yet. In addition, it is unclear what data will be recycled from previous datasets and how this point relates to the licensing of said previous dataset.

**Ethics:**

No ethical concerns.

**Limitations:**

The authors correctly point out that one of the limitations of the approach is evaluating the proposed methodology with respect to disease (e.g., when tumors are present). It would also be important to highlight that the algorithm used in the active learning step is supervised, therefore, any label biases from labels in the training datasets may potentially be propagated into the labelling of the new data.

**Opportunities For Improvement:**

It appears that the presented dataset is leveraged from existing data. In lines 50-51, “we leverage
51 existing data and incomplete labels in an assembly of 14 public datasets [43, 36]”. Therefore, the contribution seems to be the release and the improvement of labels for the public datasets, which have previously been released (see Table 2 supplementary material for listing of all the datasets). In any case, the source and permissions to release data needs to be clarified and provided.
In addition, since the active learning strategy is one of the key contributions, I expected to see an improvement after its use. However, in Table 4 (supplementary), the improvement on JHH, the improvement with respect to mDice and mNSD is typically 1-2 points, within the standard deviation for each of the 9 organs and average. Additional information about the utility of the dataset needs to be provided, and whether the provided labels are better than the existing partial labels.

Specific comments:

-	In Figure 3 (manuscript), the authors report visualizations of the sums of attention maps of each organ. The intention of the experiment and the choice of the 5% threshold was unclear.

-	It seems like the study used 3 human annotators, but the procedure of assignment of annotators was not explained. Does the active learning model improve performance of a single annotator, or all annotators. In other works, if the segmentations from all three annotators were combined (without the proposed strategy), would they be better or worse than segmentation acquired using the proposed approach?

-	Information about the training level of the annotators is missing.

-	In Table 3 (supplementary), human annotation time is recorded for each of the three annotators. In the abstract, it is mentioned that the method cuts down annotation time from 30.8 years to three weeks, however, no information is provided about timing for a specific process. It would be important to clarify time improvement of annotating a single scan with and without the proposed method and explain the offline time commitment for training and testing the model.

-	In Table 1 (manuscript), the authors report results comparing attention masks and ground truth with respect to sensitivity and precision. The choice of metrics is not clear. Why not additionally measure specificity and recall, or use metrics such as mDice and mNSD used elsewhere in the submission?

**Relation To Prior Work:**

Previous datasets and approaches seem to be well described.

**Summary And Contributions:**

The paper proposed an active learning approach to annotating segmentation masks in 8448 abdominal CT organs (and 3.2M slices). The approach is used to annotate eight organs (spleen, liver, right kidney, left kidney, stomach, gall bladder, pancreas, aorta, and postcava) in conjunction with three clinical experts. As a result, the largest to date, dataset of semantic segmentation of abdominal CT is released, several times larger than the predecessor, TotalSegmentator.

While there exist previous active learning annotation methods, it appears that active learning has been used retrospectively, rather than prospectively. Therefore, it was not used to increase efficiency in data annotation. The paper’s contribution is a novel active learning methodology and the application to construct a large annotated dataset.

---

> ### Author Response · Authors · 2023-08-21
> **Responses to Reviewer Tr7Z (1/4)**
>
> We would like to thank you for your diligent efforts and insightful comments on our paper, which have helped us think more deeply. Thanks for recognizing our “novel active learning methodology,” “the paper is well written and easy to follow,” and “such a dataset will be very useful…”. In the following, we have provided a point-by-point response to all questions raised.
>
> Q1: *The contribution seems to be the release and the improvement of labels for the public datasets, which have previously been released. It does not appear that this is a novel dataset, rather, a release of improved data labels on existing datasets. It was not immediately clear to me that the proposed labels offered more accurate information over existing labels.*
>
> A1: It is true that we did not introduce new CT volumes. After summarizing the existing public datasets (see Table 3), we found that a combination of these datasets is fairly large (a total of 8,000 CT volumes and many more are coming over the years). The main challenge, however, is the absence of comprehensive annotations. In fact, scaling up annotations is much harder than scaling up CT volumes due to the limited time of expert radiologists. Our contribution is, therefore, significant, covering two major perspectives.
>
> 1. **More comprehensive annotations.** We provide per-pixel annotations for eight organs across over 8000 CT volumes, encompassing various phases and diverse populations. The existing labels of the public datasets are partial and incomplete, e.g., LiTS only had labels for the liver and liver tumors, and KiTS only had labels for the kidneys and kidney tumors. Our AbdomenAtlas-8K offered detailed annotations for all eight organs within each CT volume. As praised by reviewer nhxF: *“As currently the largest dataset, the work significantly contributed to the field of abdominal CT imaging and can serve as the basis of the development of effective AI algorithms in the future.”*
>
> 2. **A fast procedure to annotate CT volumes at scale.** Our active learning procedure enables us to accomplish the task within three weeks, a remarkable contrast to the conventional manual annotation approach, which typically requires about 30.8 years. This accelerates the annotation process by an impressive factor of 533. The majority of the annotation workload is managed solely by only one radiologist. Furthermore, this strategy also allows us to efficiently annotate a wider range of organs and tumors in the future.
>
> 3. [Minor] **Novel dataset.** We have applied our active learning procedure to creating a large private dataset (JHH) of 5,281 novel CT volumes with per-voxel annotations. However, at the moment, we cannot guarantee that this dataset can be released soon due to numerous regulatory constraints. We are actively working with Johns Hopkins Hospital to expedite its potential release. This has been clarified in our introduction: “we commit to releasing 3,410 of the 8,448 CT volumes.”
>
> Q2: *In any case, the source and permissions to release data needs to be clarified and provided. Information of the data reuse from previous datasets need to be clarified.*
>
> A2: Thanks for your suggestion. Due to the page limit, we have now elaborated on the source and permissions in Table 3 (supplementary). To clarify, we will only disseminate the annotations of the CT volumes, whereas the corresponding CT volumes can be downloaded from their original websites. We have consulted with the lawyers at Johns Hopkins University, confirming the permissions of distributing the annotations based on the license of each dataset (summarized below). We will further include this information on our GitHub page and dataset download page, ensuring clarity for users upon the dataset usage.
>
> |  dataset name  | # of CT volumes  | # of annotated organs | use of active learning | license |
> |  ----  | ----  |  ----  | ----  | ----  |
> | AMOS | 500 | 15 | No | CC BY 4.0 |
> | AbdomenCT-1K | 1,112 | 4 | No | CC BY 4.0 |
> | TotalSegmentator | 1,204 | 104 | No | CC BY 4.0 |
> | AbdomenAtlas-8K | 8,448 | 8 | Yes | pending |

---

> ### Author Response · Authors · 2023-08-21
> **Responses to Reviewer Tr7Z (2/4)**
>
> Q3: *In addition, since the active learning procedure is one of the key contributions, I expected to see an improvement after its use. However, in **revised** Table 5 (supplementary), the improvement on JHH, the improvement with respect to mDice and mNSD is typically 1-2 points, within the standard deviation for each of the 9 organs and average. Additional information about the utility of the dataset needs to be provided, and whether the provided labels are better than the existing partial labels.*
>
> A3: Table 5 (supplementary) only showed the improvement from Step 0 to Step 1 of the active learning procedure, instead of presenting the full improvement before and after using our active learning procedure. We appreciate your valuable feedback and have made several revisions to this table.
>
> 1. We added the improvement of Step 2 in the active learning procedure. Compared to the results in step 0, both mDSC and mNSD show substantial improvement, confirming the effectiveness of our active learning procedure.
>
> 2. We marked the revised classes at each Step. The segmentation performance of some organs were marginally improved simply because (1) these organs were hardly revised at certain Steps or (2) the segmentation performance was already very high (e.g., DSC > 90%). For those revised organs (e.g., aorta), AI models showed a significant improvement from 72.3% to 82.9%.
>
> Q4: *In Figure 3 (manuscript), the authors report visualizations of the sums of attention maps of each organ. The intention of the experiment and the choice of the 5% threshold was unclear.*
>
> A4: Thank you very much for the question. We have now revised the description and presentation of Figure 3. Here is the clarification to the question.
>
> 1. **The intention behind Figure 3** is to visualize the distribution of the attention size of each CT volume. A larger attention size implies a greater requirement for revision in various regions.  Figure 3 suggests that the attention size for most CT volumes is small, but there are several significant-sized outliers. These outliers are of high priority for revision by human experts. According to the figure, the ratio of outliers is about 5% (highlighted in red). The 5% is estimated by the plot and also related to the budget of human revision for each Step in the active learning procedure. It is essential to emphasize that roughly 5% of CT volumes within each dataset are highly likely to contain predicted errors, requiring further revision by our annotator.
>
> 2. **The choice of the 5% threshold.** It is true that this number is empirical. The 5% is estimated based on (1) empirical observations (the number of outliers in Figure 3) and (2) the annotation budget at each Step in the active learning procedure. If there are many outliers or a limited budget, the threshold needs to be increased accordingly. If one is dealing with new datasets, this threshold is easily obtained by analyzing the attention size distribution as plotted in Figure 3.
>
> Q5: *It seems like the study used 3 human annotators, but the procedure of assignment of annotators was not explained.*
>
> A5: Our study recruited three annotators, comprising a senior radiologist with over 15-year experience and two junior radiologists with 3-year experience. The senior radiologist undertook the task of annotation revision in the active learning procedure. The junior radiologists were responsible for reviewing the completed AbdomenAtlas-8K annotations and making revisions if needed. Additionally, they conducted the inter-annotator variability analysis in Figure 8 and further recorded the time required for annotating each organ in Table 4.

---

> ### Author Response · Authors · 2023-08-21
> **Responses to Reviewer Tr7Z (3/4)**
>
> Q6: *Does the active learning model improve performance of a single annotator, or all annotators. In other works, if the segmentations from all three annotators were combined (without the proposed strategy), would they be better or worse than segmentation acquired using the proposed approach?*
>
> A6: As clarified in Q5, only one annotator was assigned to revise the annotations generated by AI models. We did not observe a substantial improvement in the annotator's performance in organ segmentation. This is because annotating organ boundaries is relatively less difficult for human experts with 15-year experience than annotating tumors. Therefore, our strategy has rather minor contributions to enhancing the annotation performance of human annotators. For organ segmentation, the pressing challenge lies in the time-consuming nature of the conventional full organ annotation process. In this regard, our strategy accelerates over 500 times compared with conventional ones.
>
> Q7: *Information about the training level of the annotators is missing.*
>
> A7: One annotator is a trained radiologist with 15 years of experience. The other two additional annotators are with 3 years of experience. Thanks for your question; we have clarified this in the revised manuscript.
>
> Q8: *In **revised** Table 4 (supplementary), human annotation time is recorded for each of the three annotators. In the abstract, it is mentioned that the method cuts down annotation time from 30.8 years to three weeks, however, no information is provided about timing for a specific process. It would be important to clarify time improvement of annotating a single scan with and without the proposed method and explain the offline time commitment for training and testing the model.*
>
> A8: Thank you for your feedback; we have now clarified the time estimation as follows and also included this in the revised paper (revised Section 3.3).
>
> 1. **Why 30.8 years?** We considered an 8-hour workday, five days a week. A trained annotator typically needs 60 minutes per organ per CT volume [[Park et al., 2020](https://www.sciencedirect.com/science/article/pii/S2211568419301391)]. Our AbdomenAtlas-8K has a total of eight organs and around 8,000 CT volumes. Therefore, annotating the entire dataset requires 60x8x8000 (minutes) / 60/8/5 = 1600 (weeks) = 30.8 (years).
>
> 2. **Why three weeks?** Using our active learning procedure, only 400 CT volumes require manual revision from the human annotator (15 minutes per volume). That is, we managed to accelerate the annotation process by a factor of 60x8/15=32 per CT volume. Therefore, we completed the entire annotation procedure within three weeks as reported in the paper. Human efforts: 400x15 (minutes) / 60/8= 12.5 (days) plus the time commitment for training and testing AI models taking approximately 8.5 (days).
>
> Q9: *In Table 1, the authors report results comparing attention masks and ground truth with respect to sensitivity and precision. The choice of metrics is not clear. Why not additionally measure specificity and recall, or use metrics such as mDice and mNSD used elsewhere in the submission?*
>
> A9: This is a good question, here is the clarification for the choice of metrics, and we have also clarified this point in the revised manuscript.
>
> 1. **Why sensitivity and precision**. We chose sensitivity and precision because Table 1 strived to evaluate an error detection task rather than a segmentation task (comparing the boundary of attention maps and error regions). Once an error is detected, it counts as a hit, otherwise, as a miss. Sensitivity and precision can measure how well the attention maps detect the real error regions and whether the errors being detected are real errors, respectively. Hence, these two metrics are appropriate.
>
> 2. **Why not specificity, recall, mDice, or mNSD?** Recall and sensitivity are the same by [definition](https://en.wikipedia.org/wiki/Sensitivity_and_specificity). Given that the non-error region (true negatives) significantly outweighs the error region, specificity approaches a value close to one (see the table below). Thereby, specificity cannot give us meaningful information on the error detection performance. Moreover, mDice and mNSD are commonly used to evaluate the similarity of two masks, but as discussed above, the similarity in the boundary between attention maps and error regions is not necessary. Following your suggestion, we computed mDice, mNSD, and specificity in the table below. You can see that they do not provide meaningful information on the error detection performance of attention maps.
>
> |  organ  | mDice(%)  | mNSD(%) | Specificity |
> |  ----  | ----  |  ----  | ----  |
> | Spleen | 1.0 | 1.5 | 0.99  |
> | Right Kidney | 10.1 | 11.8 | 0.99  |
> | Left Kidney | 12.3 | 15.2 | 0.99 |
> | Gallbladder | 11.7 | 17.5 | 0.99 |
> | Liver | 19.1 | 21.1 | 0.99  |
> | Stomach | 26.3 | 29.3 | 0.99 |
> | Aorta | 27.4 | 20.4 | 0.99 |
> | Postcava (IVC) | 18.8 | 20.8 | 0.99 |
> | Pancreas | 18.5 | 24.0 | 0.99 |

---

> ### Author Response · Authors · 2023-08-21
> **Responses to Reviewer Tr7Z (4/4)**
>
> Q10: *The authors correctly point out that one of the limitations of the approach is evaluating the proposed methodology with respect to disease (e.g., when tumors are present). It would also be important to highlight that the algorithm used in the active learning step is supervised, therefore, any label biases from labels in the training datasets may potentially be propagated into the labeling of the new data.*
>
> A10: Thank you for pointing out this critical aspect. Indeed, as discussed in our limitation section, tumor annotation is much more challenging than organ annotation. It is true that when annotators make mistakes on tumor annotations, it will also propagate the AI training and predicting along the supervised active learning procedure. As an extension of this study, we will add tumor annotations to the AbdomenAtlas-8K  dataset by addressing the challenge in two possible directions. **Firstly**, we plan to recruit more experienced radiologists to revise tumor annotations. **Secondly**, we will incorporate the radiology reports (based on biopsy results) into the human revision. These actions can reduce potential label biases and label errors from human annotators.
>
> Q11: *The data is not released yet. In addition, it is unclear what data will be recycled from previous datasets and how this point relates to the licensing of said previous dataset.*
>
> A11: The dataset will be made publicly available once the paper has been accepted. At the moment, all requisite code for AbdomenAtlas-8K creation has been released on GitHub and we have confirmed with lawyers at Johns Hopkins University for the permissions necessary for distributing the full annotations. To clarify, we will only disseminate the annotations of the CT volumes separately, and users will retrieve the original CT volumes, if needed, from the original sources (websites). Everything we intend to create and license-out will be in separate files and no modifications are necessary to the original CT volumes.
>
> **Reference**
>
> [1] Park, S., Chu, L.C., Fishman, E.K., Yuille, A.L., Vogelstein, B., Kinzler, K.W., Horton, K.M., Hruban, R.H., Zinreich, E.S., Fouladi, D.F. and Shayesteh, S., Annotated normal CT data of the abdomen for deep learning: Challenges and strategies for implementation. Diagnostic and interventional imaging 2020

---

> > ### Comment · Reviewer_Tr7Z · 2023-08-26
> > **great diversity of source CTs with a unified labelling protocol**
> >
> > Thank you for your comments. I believe that adding consistent and more detailed annotations across datasets in itself is a useful contribution, in addition to the newly added JHU data. Overall, the rebuttal addressed my concerns, and I have raised my rating in response.
> >
> > I believe that the contribution of releasing data from 26 hospitals worldwide (in comparison to existing datasets that are more limited in geographic and source diversity) is an important contribution. It would help if the authors could compare (perhaps create a table) with source countries to better visualize where data from previous datasets and propsoed AbdomenAtlas-8K come from.
> >
> > I was wondering whether the annotation procedures across hospitals or annotators may differ (e.g., due to to training levels or other factors), and how this may positively or negatively affect performance.
> >
> >
> > Some additional technical comments:
> >
> >
> > _"We marked the revised classes at each Step. The segmentation performance of some organs were marginally improved simply because (1) these organs were hardly revised at certain Steps or (2) the segmentation performance was already very high (e.g., DSC > 90%). For those revised organs (e.g., aorta), AI models showed a significant improvement from 72.3% to 82.9%."_
> >
> > You may consider the amount of revisions made by each organ, and reporting these statistics, or otherwise clarifying that some organs are harder to segment than others.
> >
> >
> > Figure 3 (Attention). I still do not quite understand the result. The y-axis is the sum of attention, while the caption talks about attention volume (please clarify whether these are the same).
> >
> > It may help augmenting Figure 3 (which explains attention), with a graphical visualization of components. I was also curious whether attention can be generated using one architecture but with different initializations at training. It seems like a useful tool to measure model uncertainty.

---

> ### Author Response · Authors · 2023-08-28
> **Responses to Reviewer Tr7Z (Second Round) (1/2)**
>
> Thanks for your encouraging reviews. We will attach our second-round revised manuscript on August 29.
>
> Q1: *It would help if the authors could compare (perhaps create a table) with source countries to better visualize where data from previous datasets and the proposed AbdomenAtlas-8K come from.*
>
> A1: We appreciate your valuable suggestion. In the updated Table 3, we have incorporated the source countries corresponding to each previous dataset. Furthermore, we have included a global map illustrating the worldwide source distribution of AbdomenAtlas-8K.
>
> Q2: *I was wondering whether the annotation procedures across hospitals or annotators may differ (e.g., due to training levels or other factors), and how this may positively or negatively affect performance.*
>
> A2: Thanks for your question. Consistency and accuracy are of utmost importance in medical data annotation. However, in the real world, variations in annotation procedures can occur across hospitals, laboratories, or individual annotators. When creating AbdomenAtlas-8K, we observed the variations in annotation procedures across 26 hospitals in our study. Two factors could cause these variations based on our observation.
>
> 1. **Different scanning body range.** An illustrative instance is the aorta, an elongated anatomical structure traversing both the thoracic and abdominal cavities. Notably, its morphology exhibits significant differences between these distinct anatomical regions. For abdominal CT scans, in general, the aortic arch in the chest region is typically not scanned. However, due to variations in the scanning range of CT scans across different hospitals, some CT scans also include and annotate the aortic arch. As a result, the standard annotation for the same organ provided by different hospitals varies significantly.
>
> 2. **Different annotation protocols across hospitals.** For example, the stomach and duodenum often have blurry boundaries, posing a challenge in distinguishing between these two organs. Different hospitals typically adhere to varying annotation protocols, which can further complicate the learning process for our AI model.
>
> **Positive impact to AI training:**
>
> - Variability to some extent could enhance the robustness of AI models. If a model is trained on a diverse set of annotations from different sources, it might be better equipped to generalize to novel data.
>
> **Negative impact to AI training:**
>
> - AI models trained on inconsistent annotations could produce unreliable or unpredictable outcomes. As a result, it might be challenging to reproduce AI predictions across different hospitals if there's significant variability in the source annotations.

---

> ### Author Response · Authors · 2023-08-28
> **Responses to Reviewer Tr7Z (Second Round) (2/2)**
>
> Q3: *You may consider the amount of revisions made by each organ, and reporting these statistics, or otherwise clarifying that some organs are harder to segment than others.*
>
> A3: Thanks for your suggestion. We have reported the amount of revisions by highlighting the two most significantly revised classes (i.e., aorta and postcava) at each step in blue in Appendix Table 5. These two organs have consistently shown a steady rise in mDSC during the active learning procedure. Specifically, the aorta increased from 72.3% to 83.7%, and the postcava improved from 76.1% to 78.6%.
>
> In addition, some organs are harder to segment than others due to two reasons.
>
> 1. **Diverse annotation protocols.** As mentioned in A2, there are often varying annotation protocols for the aorta among different hospitals. Consequently, achieving precise segmentation with our AI models becomes a challenging task. This often requires frequent revisions of aorta annotations by our annotators. The improvement of aorta annotations through the active learning procedure is illustrated in Table 5. Notably, there is a substantial increase in mDSC, from 72.3% to 83.7%, after undergoing two steps of active learning procedure.
>
> 2. **Blurry organ boundary.** Organs such as the pancreas often exhibit blurry boundaries, presenting a challenge for both AI and human annotations. Consequently, there is no observed improvement in annotation performance for such organs even after two steps of our active learning procedure.
>
> Q4: *Figure 3 (Attention). I still do not quite understand the result. The y-axis is the sum of attention, while the caption talks about attention volume (please clarify whether these are the same). It may help augmenting Figure 3 (which explains attention), with a graphical visualization of components. I was also curious whether attention can be generated using one architecture but with different initializations at training. It seems like a useful tool to measure model uncertainty.*
>
> A4: The “sum of attention” in the y-axis title and “attention size” in the caption are the same. Thank you for pointing this out. We have revised Figure 3 and its caption to make the terms consistent.
>
> 1. **Graphical visualization of components.** Sorry that we are not completely clear about the “component” that you were referring to. If you're referring to each organ in the attention map, please see Figure 2 and Appendix Figure 11 for the visualization. If this isn't what you had in mind, please provide further clarification on the “components”, and we'll be happy to make improvements to Figure 3. Many thanks!
>
> 2. **Whether attention can be generated using one architecture but with different initializations at training**. Yes, this is also a great approach to generate attention maps by using a single AI architecture with different weight initialization.

---

### Official Review · Reviewer_3MZ3 · 2023-07-23
**Review of "Annotating 8,000 Abdominal CT Volumes for Multi-Organ Segmentation in Three Weeks"**

**Rating:** 5
**Confidence:** 5
**Correctness:** YES
**Clarity:** YES

**Strengths:**

1. **Simple but Effective Idea:** The submission uses multiple pre-trained models to reduce label bias in medical imaging annotation, demonstrating an innovative yet straightforward solution.

2. **Clear Writing:** The paper is well-structured and articulates complex ideas clearly, making it accessible to a broad audience.

3. **Large-Scale Dataset:** The creation of the largest annotated multi-organ dataset to date, with 8,448 CT volumes, is a major accomplishment that can greatly benefit the medical imaging field.

**Additional Feedback:**

A

**Documentation:**

YES

**Ethics:**

YES

**Limitations:**

YES

**Opportunities For Improvement:**

The authors of this paper put forth a simple, effective idea that scales to much larger datasets. Their work expands upon the recent trend of using AI models to assist in annotation. For instance, systems like TotalSegmentor, AMOS, and AB1K have used similar concepts. However, I would appreciate a more comprehensive description and comparison in the related works section to better understand how this research extends upon these previous methods and mitigates model bias.

While this paper serves as an excellent technical report, my primary concern lies in its positioning as a dataset and benchmark track. Despite the authors' claim of their ability to quickly build large-scale medical datasets, the paper falls short of providing a clear explanation of the dataset's applications and potential for future research. The utility and scope of the dataset, how it might be employed by future researchers, and benchmarks against existing methods to outline future problems to be solved are all vital aspects that are currently missing.

These are crucial elements when presenting a dataset, and their absence significantly limits the impact of the paper. Therefore, I would encourage the authors to elaborate on these aspects to enhance the paper's value to future researchers and enrich its contribution to the field of medical image segmentation.

**Relation To Prior Work:**

Not fully

**Summary And Contributions:**

- Proposing an efficient method for medical image annotation, particularly for organ segmentation.
- Creation of the largest annotated multi-organ dataset from CT volumes. Significant reduction in annotation time from roughly 30.8 years to 3 weeks.
- Development of strategies for label bias reduction, effective error detection, and attention guidance for error correction.
- Summarization of a taxonomy of common errors made by AI algorithms and annotators, enabling continuous improvement of AI models and annotations.

---

> ### Author Response · Authors · 2023-08-20
> **Responses to Reviewer 3MZ3 (1/3)**
>
> We appreciate your helpful suggestions for our paper. In the following, we have responded to all questions point-to-point.
>
> Q1: *TotalSegmentator, AMOS, and AB1K have used similar concepts. However, I would appreciate a more comprehensive description and comparison in the related works section to better understand how this research extends upon these previous methods and mitigates model bias.*
>
> A1: TotalSegmentator, AMOS, and AB1K are high-quality and impactful datasets, while our AbdomenAtlas-8K has its own unique properties in four dimensions.
>
> 1. **A significantly larger number of annotated CT volumes.** TotalSegmentator, AMOS, and AB1K provided 1,204, 500, and 1,112 annotated CT volumes. AbdomenAtlas-8K provided 8,448 annotated CT volumes (around eight times larger).
>
> 2. **A notably greater diversity of the provided CT volumes.** The CT volumes in AbdomenAtlas-8K were collected and assembled from at least 26 different hospitals worldwide, whereas the makeup of TotalSegmentator and AMOS was sourced from a single country. Specifically, TotalSegmentator was from Switzerland (biased to the Central European population) and AMOS was from China (biased to the East Asian population). While AbdomenCT-1K was from 12 different hospitals, our AbdomenAtlas-8K presents significantly more CT volumes (8,448 vs. 1,112) and more types of annotated classes (8 vs. 4).
>
> 3. **The manual annotation time was significantly reduced.** The creation of AbdomenAtlas-8K used an effective active learning procedure, reducing the annotation time from 30.8 years to three weeks (see the revised Section 3.3 for a detailed calculation). This is an important scientific attempt to put active learning into practical use. As praised by Reviewer Tr7Z, *“it appears that active learning has been used retrospectively, rather than prospectively.”*
>
> 4. **Produce an attention map to highlight the regions to be revised.** The attention maps mentioned in our active learning procedure can accurately detect regions with a high risk of prediction errors (evidenced in Table 1). This capability enables annotators to quickly find areas that require human revision. As a result, it significantly reduces annotators' workload and annotation time by a factor of 533.
>
> Thank you very much for the valuable suggestion. We have now revised the related work section. To make it clearer, in conjunction with Figure 1, we also produced the following Table to signify the extension of this research over the previous ones.
>
> |  dataset name  | # of CT volumes  | # of annotated organs | # of hospitals | use of active learning |
> |  ----  | ----  |  ----  | ----  | ----  |
> | AMOS | 500 | 15 | 2 | No |
> | AbdomenCT-1K | 1,112 | 4 | 12 | No |
> | TotalSegmentator | 1,204 | 104 | 1 | No |
> | AbdomenAtlas-8K | 8,448 | 8 | 26 | Yes |

---

> ### Author Response · Authors · 2023-08-20
> **Responses to Reviewer 3MZ3 (2/3)**
>
> Q2: *The paper falls short of providing a clear explanation of the dataset's applications and potential for future research. The utility and scope of the dataset, how it might be employed by future researchers and benchmarks against existing methods to outline future problems to be solved are all vital aspects that are currently missing.*
>
> A2: Our AbdomenAtlas-8K is expected to exert applications/potentials in four perspectives for future research once released for public use.
>
> 1. **Benchmarking existing segmentation models.** In the submission, we benchmarked the most recent advances in medical segmentation, i.e., SwinUNETR [[Tang et al., CVPR 2022](https://openaccess.thecvf.com/content/CVPR2022/papers/Tang_Self-Supervised_Pre-Training_of_Swin_Transformers_for_3D_Medical_Image_Analysis_CVPR_2022_paper.pdf)]. Thanks for your suggestion, we have now enriched the comparison by evaluating three more models on the AbdomenAtlas-8K dataset, namely U-Net, UNETR, and SegResNet. The revised benchmark has been added to the main paper (see Table 2).
>
> 2. **A large dataset for developing medical foundation models.** Developing foundation models for healthcare has recently raised much attention. Foundation model refers to an AI model that is trained on a large dataset and can be adapted to many specific downstream applications. This requires a large-scale, fully-annotated dataset. We anticipate that our AbdomenAtlas-8K can play an important role in achieving this by enabling the model to capture complex organ patterns, variations, and features across different imaging phases, modalities, and a wide range of populations. This has been partially evidenced by our recent publication [[Liu et al., ICCV 2023](https://arxiv.org/abs/2301.00785)] and an ongoing project, showing that fully-supervised pre-training on AbdomenAtlas-8K transfers much better than existing self-supervised pre-training.
>
> 3. **The proposed strategy can scale up annotations quickly.** This strategy can be used for creating many medical datasets (across organs, diseases, and imaging modalities) or even natural imaging datasets. We are implementing our active learning procedure to re-create the natural imaging datasets previously produced by us [[He et al., ECCV 2022
> ](https://arxiv.org/abs/2112.00933); [Zhao et al., ECCV 2022](https://arxiv.org/abs/2111.14341)], yielding a considerably reduced amount of labeling efforts. Moreover, our strategy is being integrated into open-source software such as MONAI-LABEL at NVIDIA and ChimeraX at UCB/UCSF. This will make a difference in the rapid annotation of medical images in the near future.
>
> 4. **Enabling precision medicine for various downstream applications.**
> We showcased one of the most pressing applications—early detection and localization of pancreatic cancer, an extremely deadly disease, with a 5-year relative survival rate of only 12% in the United States. The AI trained on a large, private dataset at Johns Hopkins Hospital (JHH), performed arguably higher than typical radiologists [[Xia et al., medRxiv 2022](https://www.medrxiv.org/content/10.1101/2022.09.24.22280071v1)]. But this AI model and annotated dataset were inaccessible due to the many policies. Now, our paper demonstrated that using AbdomenAtlas-8K (100% made up of publicly accessible CT volumes), AI can achieve similar performance when directly tested on the JHH dataset (see Table 2). This study is a concrete demonstration of how AbdomenAtlas-8K can be used to train AI models that can be generalized to many CT volumes from novel hospitals and be adapted to address a range of clinical problems.
>
> [Follow-up Plans] Upon the dataset's release, we aim to host international competitions in 2024, addressing the challenges in multi-organ and multi-tumor segmentation at scale via platforms such as MICCAI/RSNA/Grand Challenge. Additionally, we are in the process of launching special issues about scaling datasets, annotations, and algorithms in leading medical imaging journals such as MEDIA and TMI.
>
> Finally, as acknowledged by Reviewer Tr7Z, *“since annotating medical images and release of data is typically a large burden which prevents AI model development, such a dataset will be very useful to facilitate development and training of AI for medical applications.”* We are thrilled to release this dataset to the public promptly, furthering our commitment to providing large-scale, fully-annotated datasets for academic purposes.

---

> ### Author Response · Authors · 2023-08-20
> **Responses to Reviewer 3MZ3 (3/3)**
>
> **References**
>
> [1] Tang, Y., Yang, D., Li, W., Roth, H.R., Landman, B., Xu, D., Nath, V. and Hatamizadeh, A. Self-supervised pre-training of swin transformers for 3d medical image analysis. CVPR 2022.
>
> [2] Liu, J., Zhang, Y., Chen, J.N., Xiao, J., Lu, Y., Landman, B.A., Yuan, Y., Yuille, A., Tang, Y. and Zhou, Z., 2023. Clip-driven universal model for organ segmentation and tumor detection. ICCV 2023.
>
> [3] He, J., Yang, S., Yang, S., Kortylewski, A., Yuan, X., Chen, J.N., Liu, S., Yang, C., Yu, Q. and Yuille, A. Partimagenet: A large, high-quality dataset of parts. ECCV 2022.
>
> [4] Zhao, B., Yu, S., Ma, W., Yu, M., Mei, S., Wang, A., He, J., Yuille, A. and Kortylewski, A., OOD-CV: a benchmark for robustness to out-of-distribution shifts of individual nuisances in natural images. ECCV 2022.
>
> [5] Xia, Y., Yu, Q., Chu, L., Kawamoto, S., Park, S., Liu, F., Chen, J., Zhu, Z., Li, B., Zhou, Z. and Lu, Y., The Felix project: deep networks to detect pancreatic neoplasms. medRxiv 2022.

---

### Author Response · Authors · 2023-08-20
**Responses to all Reviewers**

We would like to express our sincere gratitude to the reviewers for their valuable comments. To address all the reviewers’ concerns, we provide extensive experiments, discussions, and explanations. For ease of tracking, all new content in the revised manuscript (*which will be attached before August 22*) is highlighted in blue. In the following, we first provide responses for the two common concerns, then we provide point-to-point responses to all the comments from reviewers.

1. **Contributions and applications of AbdomenAtlas-8K.** Two contributions: A large-scale dataset of 8,448 annotated CT volumes and an active learning procedure that can quickly create many other large-scale datasets. **Firstly**, AbdomenAtlas-8K was a composite dataset that unified medical datasets from at least 26 different hospitals worldwide. In total, more than $60.6\times10^9$ voxels were annotated in AbdomenAtlas-8K in comparison with $4.3\times10^9$ voxels annotated in the existing public datasets. We scaled up the organ annotation by a factor of 15. Once released, AbdomenAtlas-8K can be used to benchmark existing segmentation models and foster medical foundation models for a range of downstream applications. **Secondly**, the proposed active learning procedure can generate an attention map to highlight the regions to be revised by radiologists, reducing the annotation time from 30.8 years to three weeks. This strategy can scale up annotations quickly for creating medical datasets or even natural imaging datasets.

2. **Source and permissions to release data.** We have now elaborated on the source and permissions in Table 3 (supplementary). To clarify, we will only disseminate the annotations of the CT volumes separately, and users will retrieve the original CT volumes, if needed, from the original sources (websites). Everything we intend to create and license-out will be in separate files and no modifications are necessary to the original CT volumes. We have consulted with the lawyers at Johns Hopkins University, confirming the permissions of distributing the annotations based on the license of each dataset. We will further include detailed download instructions on our GitHub page.

In addition, we have also taken this revision opportunity to improve the writing of our manuscripts and improve figures/tables to make the new material consistent with the rest of the paper. Finally, we have updated our GitHub page by releasing the code and model checkpoints. The AbdomenAtlas-8K dataset will be released upon the acceptance of the paper.

---

### Decision · Program_Chairs · 2023-09-22

**Decision:**

Accept (Poster)

**Comment:**

The authors have provided detailed responses addressing the reviewers' valuable feedback. I appreciate the authors' sincere efforts to improve the work based on the reviewers' suggestions.

A key concern was around the dataset release. The authors are working to enable the release of the private dataset of CT volumes. The code to generate the dataset using the proposed active learning approach has already been released.

Another question was around the generalizability of the approach to more structures and modalities. The authors have helpfully discussed explorations on expanding to 13 more organs, various tumors, and modalities like MRI. Integrating the approach into open-source tools will also aid adoption.

The comparisons between AbdomenAtlas-8K and TotalSegmentator were discussed further. The authors agreed that reducing annotations to 8 organs is a limitation but one stemming from their focus on active learning. Expanding annotations is planned. The time savings arise from prioritizing volumes needing revision versus reviewing all volumes.

Some metrics like specificity were added to address a reviewer's concern. Inter-annotator variability was also clarified - it assessed the AI model's quality rather than human bias. Overall, the authors' detailed responses and additions have satisfactorily addressed the reviewers' comments.

I recommend accepting this paper given the novelty of the large-scale annotated dataset achieved via an efficient active learning approach. This can enable training robust AI models for medical image analysis.